# Plug-and-play evolution of the *Klebsiella pneumoniae* capsule locus enables serotype exchange across genetic backgrounds

Julie Le Bris [1,2]*, Hugo Varet [3], Eduardo P. C. Rocha [1], Olaya Rendueles [1,4]

1 Microbial Evolutionary Genomics, CNRS UMR3525, Institut Pasteur, Université Paris Cité, Paris, France, 2 École Doctorale Complexité du Vivant, Collège Doctoral, Sorbonne Université, Paris, France, 3 Bioinformatics and Biostatistics Hub, Institut Pasteur, Université Paris Cité, Paris, France, 4 Laboratoire de Microbiologie et Génétique Moléculaires (LMGM), CNRS UMR5100, Centre de Biologie Intégrative (CBI), Université de Toulouse, Toulouse, France

* julie.le-bris@pasteur.fr

## Abstract

Understanding how complex, multi-gene systems evolve and function across genetic backgrounds is a central question in molecular evolution. While such systems often impose costs through epistatic interactions, some may behave as modular, "plug-and-play" units that retain function with minimal disruption. Here, we tested this using the polysaccharide capsule locus of *Klebsiella pneumoniae*, a highly exchangeable and fast-evolving locus, as a model. We genetically engineered capsule exchanges (swaps) across diverse genetic backgrounds and combined transcriptomics, fitness assays, and evolution experiments to show that capsule exchange has negligible effects on global expression and only marginal fitness costs, regardless of capsule type (or K type). Adaptation to capsule-costly environments consistently reduced capsule production regardless of K type, revealing shared adaptive trajectories rather than K type-specific pathways. Moreover, K type-specific traits involved in bacterial virulence, such as biofilm formation and hypermucoviscosity, were conserved across genetic backgrounds. This reveals that capsule swapping can directly shape host-pathogen interactions and influence within-patient evolution. Our findings provide strong evidence that capsule loci display plug-and-play dynamics: they are transferable, functional across contexts, and minimally disruptive to the host genome. This allows capsules to be seamlessly swapped, and help explain the evolutionary success, ecological versatility, and pervasive exchangeability of capsules in *K. pneumoniae*.

## Introduction

Bacteria continuously expand their functional repertoire through a variety of evolutionary mechanisms, including gene duplication and diversification [1,2], domain rearrangement, and horizontal gene transfer. These processes can result in acquisition

**Data availability statement:** All RNA-seq FASTQ files are available in the GEO database (accession number: GSE306874). All raw DNA sequence reads are available from the BioProject database (accession number: PRJNA1365496).

**Funding:** This research was partly financed by the Georges, Jacques, and Elias Canetti prize awarded to O.R https://research.pasteur.fr/en/call/2025-georges-jacques-and-elias-canetti-prize-call-for-nomination/. This work was partially supported by the grant from ANR (Agence nationale de recherche) [ANR 22 CE20 00181 BETinCAP] awarded to O.R https://anr.fr/Projet-ANR-22-CE20-0018. J.L.B is supported by a doctoral contract from the French Ministry of Higher Education and Research (MESR) and the ANR "Chaire d'Excellence en Biologie-Santé" [ANR-24-CHBS-0006 microINTERACT] awarded to O.R https://anr.fr/fr/detail/call/chaire-dexcellence-en-biologiesante-appel-a-projets-2023/. The funders had no role in study design, execution, interpretation, or writing of the study.

**Competing interests:** The authors have declared that no competing interests exist.

**Abbreviations:** AUC, area under the curve; DGE, differentially expressed genes; FACS, fluorescence-activated cell sorting; FLP, flippase; HMV, hypermucoviscosity; *Kpn*, Klebsiella pneumoniae; LB, Luria-Bertani (nutrient-rich media); M02, M63B1 supplemented with 0.2% glucose (nutrient-poor media); PCA, Principal Component Analyses.

of analogous functions or in novel ones, amongst which, antibiotic resistance, virulence, or metabolic pathways [3,4]. While numerous studies have documented the advantages of such gains in the laboratory and natural populations, their evolutionary success is thought to be rare [5,6]. The integration of novel genetic systems into existing cellular networks may not be seamless and incur in fitness cost due to regulatory conflicts, metabolic imbalances, or structural incompatibilities [7]. Their success depends on the gene-by-environment fitness effect [8] and on complex epistatic interactions with the host genome [9], where the impact of a gene is shaped by the recipient genetic background. Indeed, epistasis is widespread in nature, influencing evolutionary dynamics in cancers, viruses [10], and bacteria [11,12]. It can be positive, for instance, large plasmids can promote the maintenance of other, smaller plasmids in bacterial populations [13], but it can also be negative [12,14], requiring compensatory mutations to limit the fitness costs [15,16]. These interactions underscore the context-dependency of genetic innovation and the challenges of functional integration.

Promiscuous systems which are frequently mobilized across genetic backgrounds are hypothesized to follow a plug-and-play evolutionary dynamic, whereby successful integration results in minimal epistatic interference with the host genome [17, 18]. Such systems are characterized by a high degree of modularity, that is, their functionality is largely self-contained and does not rely on tight interactions with host-specific pathways. This architecture allows expression and maintenance of the novel function with limited physiological burden or need for compensatory mutations. The plug-and-play model has been extensively documented in phage biology and protein domain shuffling [19, 20], where modular elements maintain functionality across hosts. More recently, sedentary chromosomal integrons were described as genetically and functionally isolated units where cassettes are integrated and expressed in a plug-and-play manner [21]. Modularity enhances evolvability by enabling the reconfiguration of cellular functions through the acquisition or exchange of discrete, low-conflict genetic modules in response to changing environmental conditions [22, 23]. Despite its conceptual appeal, whether true plug-and-play dynamics apply to large, multi-gene chromosomal systems in bacteria remains untested. Notably, a key question is whether such systems impose universal fitness trade-offs upon transfer or whether their architecture allows for broad functional compatibility with minimal host rewiring.

An example of a complex multi-gene system is the extracellular polysaccharide capsule locus (*cps*). It is one of the fastest-evolving loci in Bacteria due to horizontal gene transfer and characterized by elevated recombination rates [24,25]. Present in many facultative pathogens and widespread in environmental species [26], this highly diverse surface structure is a major virulence factor with critical roles in microbial ecology and evolution. Its impact further extends to clinical settings, biotechnology, and public health [27]. Group I capsules, the most prevalent, also known as Wzx/Wzy-dependent capsules, contain both conserved core genes and a variable region encoding oligosaccharide modifying and polymerization enzymes. This variable region results in different genomically characterized capsule locus types, which often lead to different serum reactivities, mostly linked to a difference in biochemical

compositions. Hereafter, we will refer to these different capsule locus types as K types, which were assigned based on sequence information rather than classical serological typing [28,29]. Capsule locus diversity is shaped by strong diversifying selection [28, 29], likely driven by the immune system, phage, and/or protist predation [27–29].

In the enterobacteria *Klebsiella pneumoniae* (*Kpn*), an opportunistic pathogen able to colonize a broad range of environments [30,31], specific capsule types have been linked to distinct diseases, such as K3 with rhinoscleromatosis [32] and K2 with inflammatory bowel diseases [33,34]. Recent findings suggest that K type influences *Kpn* bloodstream survival rates [35], pointing to K type-specific contributions to virulence. In *Kpn*, over 140 different K types have been described [36]. The current model posits that the whole capsule locus is transferred horizontally, most likely by conjugative elements. Once in the recipient cell, the capsule integrates in a one-step large recombination event spanning the whole conserved locus known as the K-locus (10–30 kb) [25]. These exchanges occur preferentially across genomes encoding biochemically similar capsules, rather than by genetic relatedness [25]. Whereas successful swaps typically involved capsules sharing a significant proportion of sugar moieties (an average of 2.4), similarities in gene repertoires were poor predictors of their eco-evolutionary success. This suggests that strong epistatic interactions occur between host genome and the K-locus [25]. This presents an interesting evolutionary conundrum, as the pervasiveness of capsule exchanges suggests this process occurs by a plug-and-play process in which there is minimal disruption to the cell, e.g., low negative epistasis. Furthermore, the conserved core genes of the locus are more integrated in the cell metabolism, and their exchange may have a stronger impact on host fitness [37], whereas the noncore (accessory), K type-specific genes, are expected to have low levels of negative epistasis as they have not co-evolved with the host genome. Finally, capsule exchanges also reshape the surface glycobiology which can impact fitness by modifying metabolic requirements, envelope properties and cellular interactions with the environment [38–40].

Several studies have engineered *in vitro* capsule locus swaps in *Kpn*, primarily to address the inheritance of virulence alongside capsule type [35,41,42]. Yet, the evolutionary mechanisms driving swaps, alongside the broader evolutionary consequences of capsule exchange, such as its effect on bacterial fitness remain unexplored. In *Streptococcus pneumoniae* (*Spn*), capsule-switched mutants engineered *in vitro* displayed distinct growth patterns that were inherited with the capsule operon type [43]. We hypothesized that if there were any transcriptional changes or fitness costs, these should occur in capsule exchanges between the most biochemically different K types. To test this, we expanded a collection of strains in which different capsule loci were introduced in different genetic backgrounds [44], and leveraged transcriptomic profiling with large-scale pairwise competitions and evolution experiments. This integrative framework allowed us to directly quantify both regulatory and fitness consequences of K type exchange and compare the effects across K types at different steps of host-pathogen interactions. Our results show there is little, if any, impact of K type exchange in the regulatory network of the cell and only marginal metabolic or energetic cost. Our study provides a clear example of plug-and-play in Bacteria and demonstrates that the widespread transferability of the capsule locus across *Klebsiella* species and beyond is underpinned by true modularity.

## Results

### Genetic background is the primary determinant of capsule production

To explore potential fitness effects and epistatic interactions between the capsule locus and its host genome upon integration by horizontal gene transfer, we expanded a collection of strains to encompass five native strains, their five respective acapsular mutants (Δcap) alongside 19 capsule-swapped strains. Specifically, five phylogenetically diverse genetic backgrounds—three hypervirulent (*Kpn* NTUH K2044, *Kpn* BJ1, and *Kpn* CIP 52.145), one commensal (*Kpn* ST45) and one environmental (*K. variicola* -*Kva*- OM26) strain (Table 1)—expressed one of four clinically relevant and well-characterized K types (S1 Fig) associated to hypervirulence (K1 and K2), rhinoscleromatitis (K3) [32], and carbapenem-resistance (K24). Of note, K3 could not be introduced into a *Kva* OM26 genetic background. These 19 capsule-swapped strains include the control strains, whereby each strain's native capsule type was reintroduced into the Δcap mutant. In two

**Table 1. Genetic backgrounds and K types used in this study to generate all the capsule-swapped strains. The identities of the different native capsule loci compared to those reintroduced as controls are included.**

**NATIVE STRAINS**

| Strain | Native K type | O-antigen type | Sequence Type | Origin |
|---|---|---|---|---|
| *Kpn* NTUH K2044 | K1 | O1αβ,2β | ST23 | Human liver abscess, Taiwan [45] |
| *Kpn* BJ1 | K2 | O1αβ,2α | ST380 | Human liver abscess, France [46] |
| *Kpn* CIP 52.145 | K2 | O1αβ,2α | ST66 | Virulent, K2 reference strain, Indonesia [46] |
| *Kpn* ST45 | K24 | O2α | ST45 | Human carriage, feces, the Netherlands [46] |
| *K. variicola* OM26 | K64 | O3αβ | | Environmental, France, CIP 80.47 |

**K TYPES**

| Capsule locus | Strain of origin | Other variants and % of identity |
|---|---|---|
| K1 | *Kpn* SA12 | K1_NTUH K2044; 99.98% identity (SNPs including in *wzc*) |
| K2 | *Kpn* BJ1 | K2_CIP 52.145; 98.24% identity (truncated *oatWY*) |
| K3 | *Kpn* ATCC13883T | K3 reference strain |
| K24 | *Kpn* ST45 | |

genetic backgrounds (*Kpn* NTUH K2044 and *Kpn* CIP 52.145), complementation involved an alternative capsule locus of the same K type, which exhibited minor genomic differences, and resulted in minor, but significant, differences in capsule production (Table 1, S1 and S2A Figs). These strains will be collectively referred to as capsule-swapped strains. Their genetic and biochemical similarities are portrayed in Fig 1A.

We first verified that all capsule-swapped strains produced detectable levels of extracellular capsule (Fig 1B, 1C), and that K type complementation would result in similar levels of capsule production to the wild type strain (S2A Fig). Capsule quantification revealed that the native K type was not necessarily the most expressed in the original genetic background (Fig 1C), and that different strains produce different amounts of capsule (Fig 1B, multifactorial ANOVA, $p < 2e{-}16$, S1 Table).

The capsule locus contains three promoter regions [49] (S1 Fig), while *ca.* 100 recently identified capsule regulators distributed across the genome [39,50] can further enhance or repress capsule transcription. These multiple regulatory layers raise the question of whether capsule production is driven more by the genetic background or the K type. A step-wise linear regression model reveals that the genetic background is the main factor explaining variation in capsule production. Interestingly, across all K types, capsule production followed the same trend: *Kpn* CIP 52.145 > *Kpn* BJ1 > *Kpn* NTUH K2044 > *Kpn* ST45 > *Kva* OM26 (Figs 1B and S2B). Also, independently of the genetic background, K types were associated with different levels of capsule production with the ranking: K1 > K2 > K24 > K3 (Figs 1C and S2C). Our data show that capsule production follows a strong hierarchical pattern dictated by the genetic background and the K type. Specifically, certain genetic backgrounds consistently produce a higher level of capsule regardless of K type, while some K types are similarly expressed across diverse genetic contexts. Furthermore, the interaction between genetic background and K type in our regression model resulted in a low F-value (F = 7.9), indicating limited interplay between these two factors. Finally, capsule production measured by the uronic acid method is positively correlated to capsule volume [44] and its thickness (S3 Fig). Collectively, our data suggests that intrinsic properties of capsule loci are conserved among strains and that the variance observed is mostly explained by independent and additive effects of the K type and the genetic background.

## Newly acquired capsule loci result in minimal gene expression changes

The adaptive response to a novel genetic element or function can occur either through transcriptional changes or genetic modifications. While the latter are difficult to revert [51,52], transcriptional response is quick, involves no lasting costs, and is not passed on to future generations [53]. We hypothesized that new K types could alter surface properties –causing

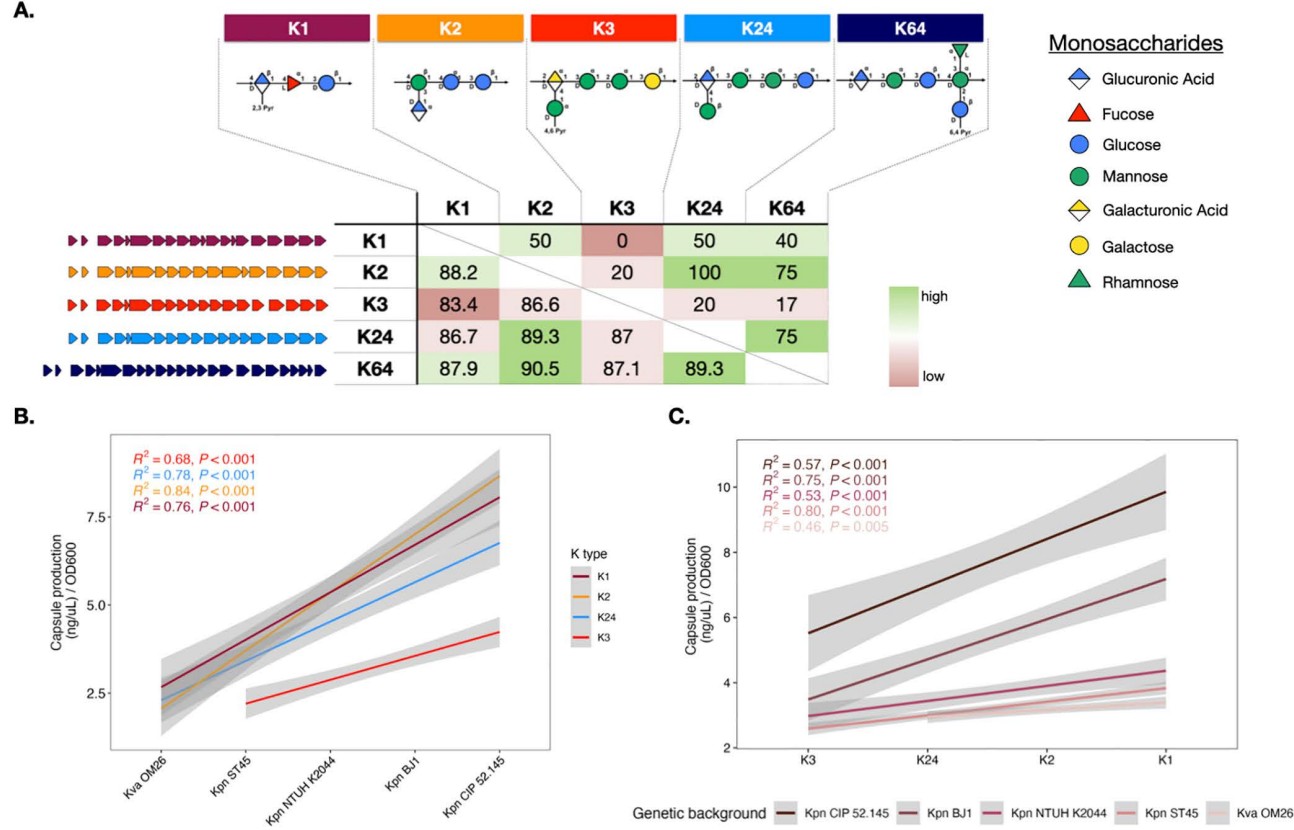

**Fig 1. Characteristics and capsule production of capsule-swapped strains. A.** Genetic (lower triangle) and biochemical (upper triangle) relatedness between the different capsule types analyzed in this study. Biochemical similarity was calculated as the percentage of residues shared between two K types (# of shared unique oligosaccharides / # of different non redundant oligosaccharides). Oligosaccharidic compositions and models were taken from K-PAM [47]. Genetic relatedness was calculated taking into account all homologous genes between two K types (protein by protein alignment, bidirectional identity averaged by pairwise comparison across proteins, see Methods). Monosaccharides composing the different capsules are depicted according to the Symbol Nomenclature for Glycans convention [48]. **B** and **C.** Capsule production of each capsule-swapped strain was determined by the uronic acid method, which directly correlates with cell volume as previously shown in Haudiquet and colleagues [44], and thickness (S3 Fig). The two graphs are generated from the same dataset with either genetic background (B) or K type (C) indicated on the x-axis. Different colored lines represent linear regressions for each K type (B) or genetic background (C). Capsule production in all genetic backgrounds is as follows; K3 < K24 < K2 < K1, with the exception of CIP 52.145 where K3 < K24 < K1 = K2. Capsule production of the native strains is shown in S2A Fig and all raw data for each strain are presented in S2B, S2C Fig. The data underlying this Figure can be found in S1 Data.

protein jamming, electrostatic shifts, and membrane disorganization—especially with decreasing biochemical relatedness between the native K type and the new one (Fig 1A). Also, we expected that gene expression changes following capsule exchange will be more similar between K types with greater biochemical relatedness. To study this, we performed an RNA-seq analysis of capsule-swapped strains during exponential phase in nutrient-poor medium where the capsule is produced at higher levels [31]. Acapsulated mutants of each genetic background were also sequenced and used as controls (see Materials and methods).

We first investigated the expression of the different capsule types across the different backgrounds compared to their native capsule (see Materials and methods). Among the 194 capsule genes analyzed, across all K type and genetic background combinations, only 18 genes were differentially expressed (nine upregulated and nine downregulated) (S4 Fig). We performed a Principal Component Analyses (PCA) (see Materials and methods) to test if each K type is associated with a specific transcriptional profile when expressed in different genetic backgrounds. Capsule types are separated

by the first two components, although only very mildly so, with the exception of K3 (S5A–S5B Fig). This indicates that the gene expression pattern of K3 capsule is different from the others. Indeed, K3 is the most distinct K type in the set, sharing the fewest number of sugar residues with other K types (either one or none) (Fig 1A). At the gene level, very few commonalities in terms of gene expression patterns are observed, if any. In *Kpn* ST45, the first enzyme of the capsule biosynthesis pathway (*wcaJ*) is significantly downregulated upon introduction of any new K type (Fig 2A). Yet, the same was not observed in other genetic backgrounds. Furthermore, some genes like *wza*, *wzb*, and *wzc* are downregulated in ST45-K3 whereas *wza* is upregulated in OM26-K1.

We next tested whether the introduction of a novel K type affects gene expression beyond the capsule locus. PCA analyses of the first two components did not allow the separation by K type, indicating that the transcription profiles in a given genetic background were very similar and independent of the expressed K type (S5C Fig). In most strains, we observe that only a small subset of the gene repertoire ($\bar{x} = 1.4\%$, median $= 0.36\%$, but reaching 8.8%, depending on the background), has a significantly altered expression, but these changes are marginal in magnitude (adjusted $P < 0.05$ with $|\log_2$ fold-change$| < 1$) (Fig 2B, S3 Table). Only very few genes were largely differentially expressed (adjusted $P < 0.05$ with $|\log_2$ fold-change$| > 1$) (Fig 2C). Contrary to our expectations, there were no changes in proteins associated to membrane biogenesis or homeostasis or in the core metabolism, or in any other pathway (Fig 2C). Only in CIP 52.145, the integration of the K1 capsule locus resulted in a significantly higher number of differentially expressed genes (169 genes)—mostly upregulated—compared to any other combination. Three main KEGG pathways were enriched: starch and sucrose

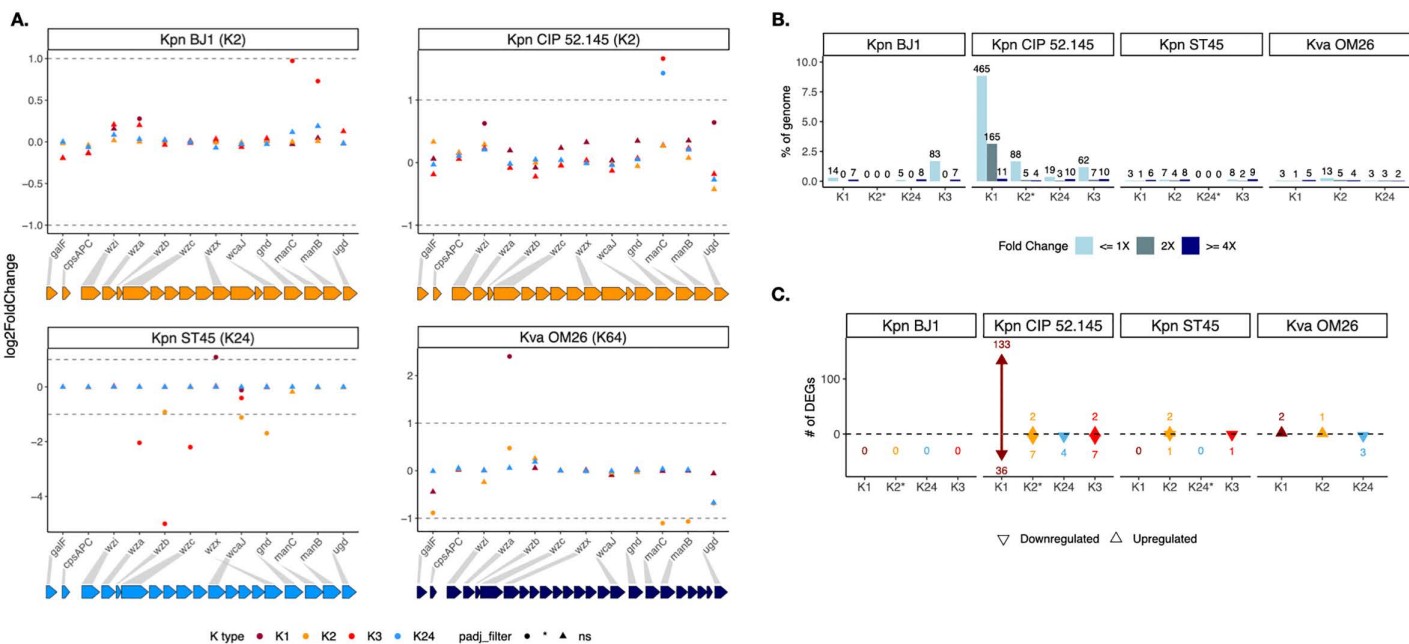

**Fig 2. Newly acquired capsule loci result in minimal gene expression changes. A–C.** Differential gene expression is analyzed within the given genetic background (indicated in the upper frame). A. Log$_2$FC (fold change) of core capsule locus genes, i.e., genes common to all the K types. Shape indicates the significance of the adjusted *p*-value and the color represents the capsule type. The genetic background is indicated at the top of each sub-panel. B. Percentage of differentially expressed genes (DEGs, adjusted $P < 0.05$) in each genetic background when considering all genes (including the capsule locus). DEGs are subdivided according to their respective Fold Change indicated by the color (corresponding to $|\log_2$ fold-change$| > 0$, 1, or 2). The corresponding raw number of DEGs is indicated at the top of the bar. C. Number of DEGs (adjusted $P < 0.05$ with $|\log_2$ fold-change$| > 1$) per capsule swap in each genetic background, either up or down regulated when considering all genes except the capsule locus. Asterisk beside K loci (*) indicates the native K type of each strain. The data underlying this Figure can be found in S1 Data.

metabolism, providing precursors for capsule polysaccharide synthesis; oxocarboxylic acid metabolism, closely linked to central carbon metabolism; and arginine biosynthesis, recently implicated in regulating the hypermucoviscosity phenotype of *Klebsiella* [54]. Genes associated to lysin degradation were also upregulated, whereas the *mrk* operon, encoding type 3 fimbriae, was downregulated.

Collectively, our results indicate that capsule expression is constant independently of the genomic context, as suggested by the capsule production quantifications mentioned above. Furthermore, despite the changes on the surface imposed by radically different glycobiology of the different K types studied here, our analyses show that introduction of novel K types does not result in a major cellular transcriptional rewiring.

## Capsule exchanges impose minimal fitness costs and reveal capsule type-dependent transitive fitness hierarchies

Given the few transcriptional changes observed, and their low magnitude, we hypothesized that K type exchanges could follow a plug-and-play dynamic, resulting in low/no detectable fitness burden. To test this, we first measured growth in nutrient-rich (LB) medium, in which capsules were shown to be costly, and in nutrient-poor (M02) medium in which capsules provide a fitness advantage [31]. We compared the capsule-swapped strains to the native K type by calculating the area under the curve (AUC) which takes into account lag time, generation time, and maximum yield. Our results show that strains with new K types do not grow significantly less, independently of the environment (Figs 3A and S6A, S6B). Across all strain x environment growth tests (*N* = 38), in only five instances, capsule-swapped strains grew significantly less, whereas in 11 instances they grew significantly better (S6A, S6B Fig). Our data reveal a negative correlation between the amount of capsule produced in nutrient-rich medium and growth (Fig 3B). Specifically, the replacement of the large K1 capsule by any other K type resulted in significant growth benefits in rich medium (S6A, S6B Fig). Using a stepwise linear regression model, we show that when the capsule is more costly (LB), both the genetic background first, and then the K type drive growth (S2 Table). Yet when the capsule, irrespective of the K type, is advantageous, growth was primarily governed by the genetic background. Finally, as observed for capsule production, there is little interaction between K type and genetic background (S1 and S2 Tables).

To further assess the K type effect on the fitness of the host, we performed fluorescent-based *in vitro* competition assay in nutrient-poor medium. We chromosomally tagged the strains at the attachment site of the transposon Tn7 (attTn7) with red fluorescent (mCherry) protein coding gene. We performed all pairwise competitions in all four *Kpn* genetic backgrounds (NTUH K2044, BJ1, CIP 52.145, and ST45) (*N* = 336 competitions). We first confirmed that the cost of the fluorescent protein expression was negligeable and that the reinsertion of the native K type in acapsular mutants did not alter fitness compared to the native strain (S6C Fig). We then tested whether in direct competition with the native K type, a new K type had reduced fitness. Surprisingly, we found no significative differences (Fig 3C). More interestingly, our data show that fitness of capsule-swapped strains is mainly dictated by the K type (multifactorial ANOVA, $p < 2e{-}16$, S1 Table). The relative fitness shows a gradual increase from the acapsular mutant to K1 (Fig 3D). Statistical analysis of the fitness hierarchy using Kendall's correlation rank confirmed such transitive relation ($\tau = 0.791$, $p = 5.3644e{-}6$). Furthermore, stepwise linear regression ranked the K types as follows: K3 < K24 < K2 < K1 with K1 being the fitter K type in nutrient-poor medium (S2 Table). Such ranking inversely mirrors the cost during growth in nutrient-rich environments, where the capsule is selected against. Of note, a slight cost was observed for *mCherry* integration in the K1 swap of *Kpn* CIP 52.145, the fittest capsule-swapped strain, suggesting its fitness advantage may be underestimated (S6C Fig).

Altogether, our findings reveal that while acquisition of a novel capsule type does not necessarily result in an additional cost for the native strain, the K type still influences fitness, with thicker capsules—such as K1 [44]—conferring a selective advantage over thinner types (K2, K24, and K3) [44] in nutrient-poor environments.

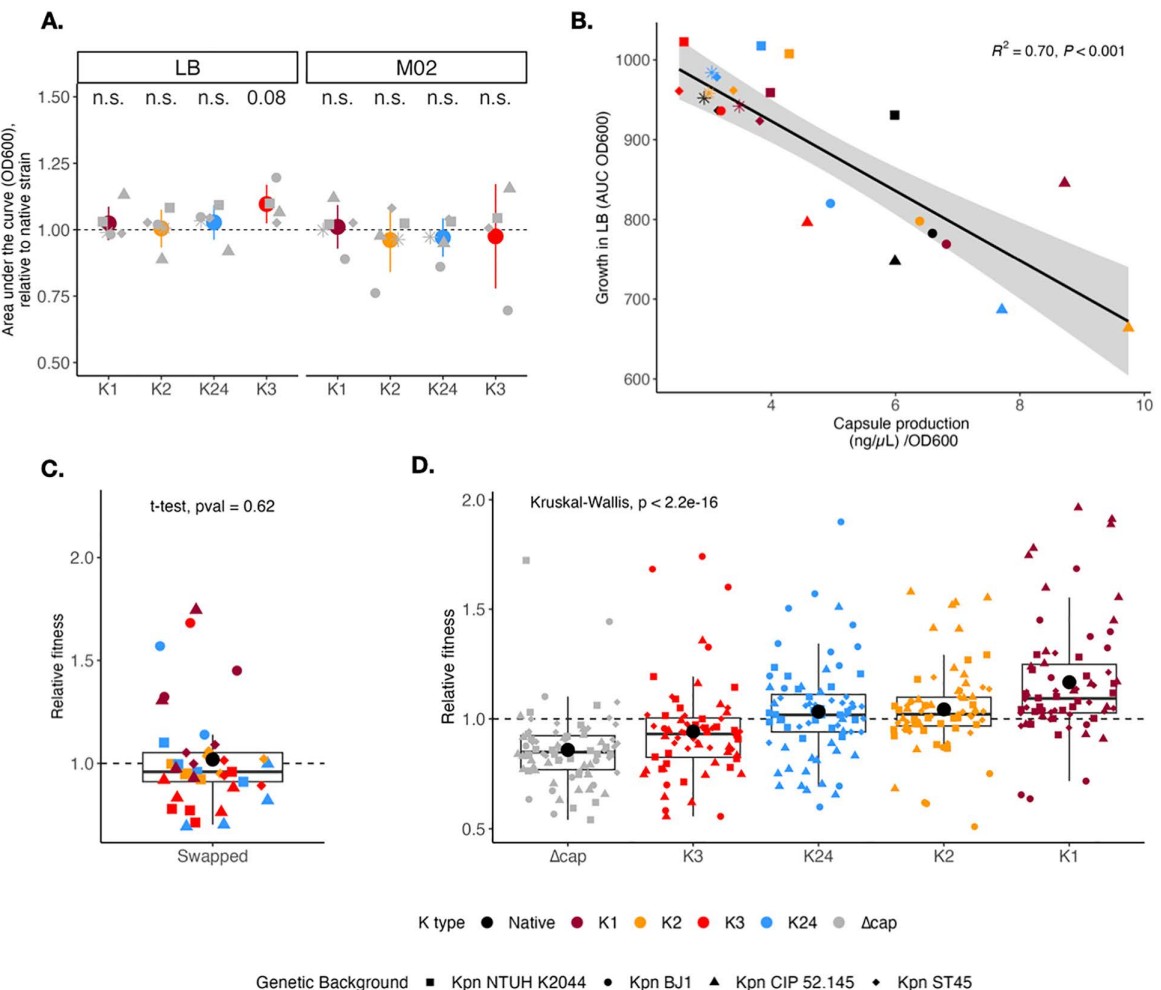

**Fig 3. Growth and fitness effect of inserting a novel capsule type across genetic backgrounds. A.** Comparison of the area under the growth curve (AUC $OD_{600}$) between native strains (dashed line) and capsule-swapped strains. AUC was estimated by the formula *trapz* from the pracma package in R. Each point represents the mean of at least three independent biological replicates. Individual error bars are not depicted for visibility purposes. **B.** Correlation between the amount of capsule produced and the growth (area under the curve, AUC) of each capsule-swapped strain in rich medium. The shape of the points corresponds to the genetic background and color to the K type. Each point represents the mean of at least three independent biological replicates. Error bars are not included for visibility purposes. Black line represents a linear regression ($R^2$). **C.** Fitness of capsule-swapped strains in direct competition with the native strain, as measured by flow cytometry after 24 hours of coculture in poor medium. *P*-value corresponds to one-sample *t* test, difference from 1. **D.** Relative fitness of capsule swap strains grouped by K type, in poor medium. *P*-value corresponds to Kruskal–Wallis means comparison test. The data underlying this Figure can be found in S1 Data.

## Thickness and common genetic pathways drive capsule inactivation of native and novel K types

We expect that different capsule types impact evolutionary trajectories in distinct ways, leading to K type-specific adaptive patterns across environments that impose varying costs on capsule production. Indeed, previous research shows that in rich media, acapsular clones rapidly emerge within 20 generations, but in poor media, all clones retain their capsules [31]. If there is a fitness cost associated to the newly acquired capsule, we would expect acapsular mutants to be more strongly selected and emerge faster compared to the native strain. To test this, we performed a short evolution experiment by transferring daily all strains for 15 days in both rich (LB) and poor (M02) media. This accounts for *ca.* 100 generations. We visually characterized and counted colonies from all populations and scored the percentage of acapsular clones every

day. In line with our previous experiments, in nutrient-poor media, no capsule inactivation was observed regardless of the K type and the genetic background (S7A Fig).

On the contrary, in nutrient-rich medium, the proportion of capsulated clones rapidly decreased in all the capsule-swapped strains during the first 10 days (Fig 4A). However, emergence of acapsular clones was not faster in the swapped strains compared to their native strain, as revealed by the area under the curve of capsule inactivation (Fig 4B). A closer inspection of the curves revealed that, during the first 5 days, capsule inactivation was faster in populations with K1 and K2 capsule types expressing thicker capsules compared to K3 and K24 swaps (Figs 4C and S7B). This is in line with the K type ranking observed when considering their capsule production and thickness (Figs 1 and S3), and inversely mirrors the fitness advantages provided in nutrient-poor medium (Fig 3D).

Towards the end of the experiment, capsulated clones increased in several populations of NTUH K2044, ST45, and BJ1 carrying either native or swapped K types, and reaching up to ~50% in frequency (Fig 4A). These clones produced significantly less capsule than their ancestors (Fig 4D), suggesting a common compensatory response to the cost of capsule production [39]. Genome sequencing of intermediate clones revealed frequent *wcaJ* mutations in ST45 and

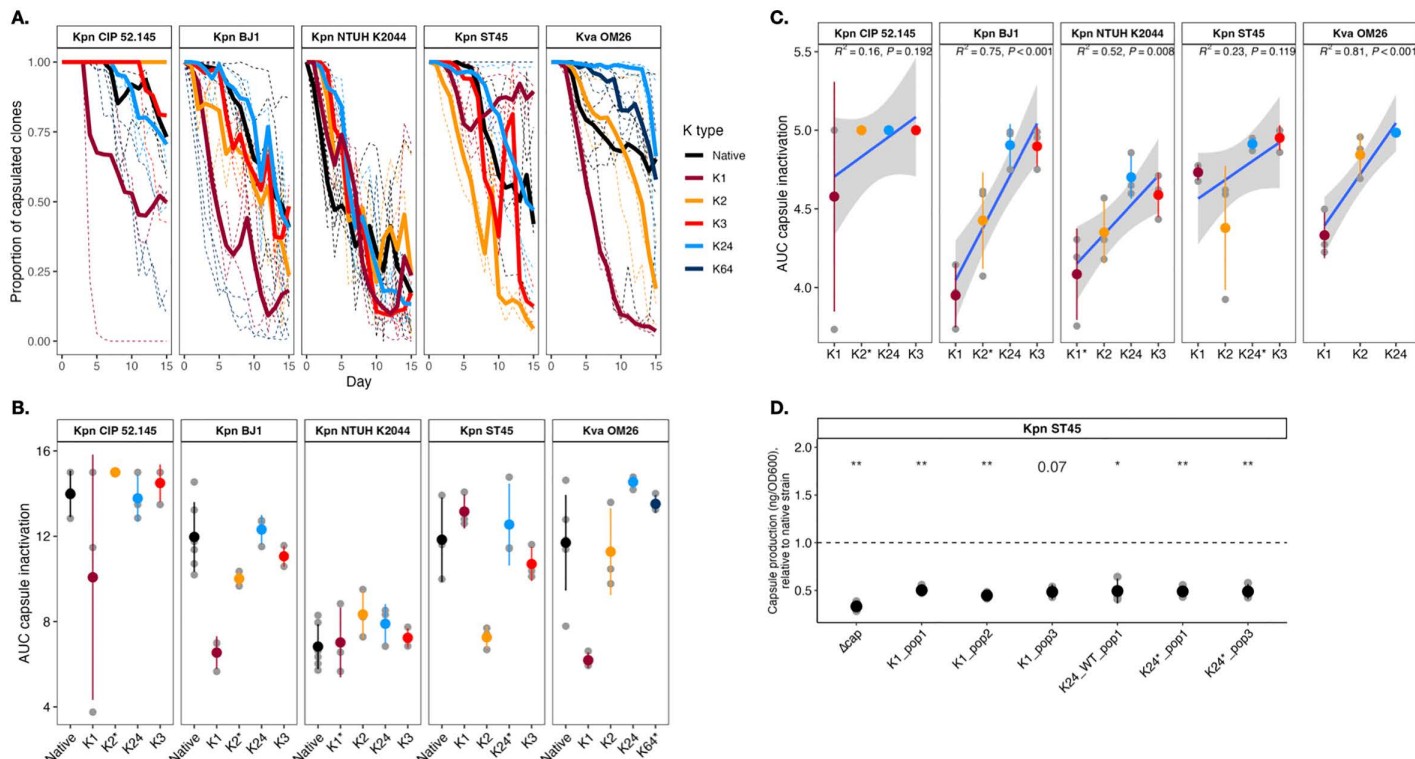

**Fig 4. Experimental evolution of capsule-swapped strains. A.** Proportion of capsulated clones through time after daily transfers in nutrient-rich medium. Independent replicates are indicated in dashed lines and bold lines correspond to the means of at least 3 independently-evolving populations. Evolution in nutrient-poor medium is shown in S7A Fig. **B.** Area under the curve (AUC) of the proportion of capsulated clones during the 15 days of the experiment in nutrient-rich medium. Gray dots correspond to independent populations and the colored dot indicates the mean. The native strain is shown in black. **C.** Linear regression ($R^2$) between capsule K type and area under the curve calculated from the first five days of the evolution experiment. **D.** Capsule production of intermediately capsulated clones, derived from different capsule-swapped ST45. Capsule production is portrayed as relative to their respective non-evolved (ancestral) strain. On the x-axis, the K type is followed by the number of the population/replicate from which the clone was isolated. Each black point represents the mean of at least three independent biological replicates. *$p < 0.05$; **$p < 0.01$; ***$p < 0.001$ one-sample *t* test. Capsule quantification of the ST45 acapsular mutant (Δcap) is included as comparison. All capsulated clones have a mutation in Tat translocation system. Similar observations were made in other genetic backgrounds and with other K types. The data underlying this Figure can be found in S1 Data.

either plasmid loss or *rmpA* mutations in NTUH K2044, both known to reduce capsulation [25,52]. Unexpectedly, both strains also showed parallel mutations in the *tat* operon, including a nonsynonymous SNP in *tatC* (S4 and S5 Tables), a translocator of folded proteins from the cytoplasmic to periplasmic space, as well as in other genes (*zipA*, *envC*). These mutations, along with the absence of some TatC substrates in the periplasmic space, are expected to impair proper cell segmentation after cell division, leading to cell-chain formation, as shown previously [55].

Our data indicate that adaptation to an environment in which capsules are costly follows the same dynamics in native and swapped strains, suggesting that mitigating the generic cost of capsule production is larger than the cost imposed by a specific and novel K type.

## Capsule type can alter virulence-associated phenotypes whereas resistance to biotic stress is dependent on the genetic background

One of the hallmarks of the plug-and-play model is that the newly inserted element should retain the same function across different genetic backgrounds. To address this, and given the importance of the capsule in *Kpn* epidemiology and patho-genicity, we tested different virulence-associated traits across the capsule-swapped strains. More specifically, we eval-uated strain performance across representative life stages of *Kpn*: survival through the gastro-intestinal tract—including resistance to bile salts and oxidative stress, which can disrupt bacterial membrane [56], colonization ability, (i.e., biofilm formation), and hypermucoviscosity, which is associated to hypervirulence [39].

We first tested the role of capsule type in resistance to physiological (0.05%) and elevated (0.5%) concentrations of pri-mary (sodium cholate) and secondary (sodium deoxycholate) bile salts (Figs 5A and S8A), and to oxidative stress, using 5 and 10 mM of hydroxide peroxide (Figs 5B and S8B, S8C). Insertion of any given capsule type did not result in differences in resistance to physiological (0.05%) concentrations of secondary bile salts (Fig 5A), or resistance to oxidative stress (10 mM) (Fig 5B).

Using a microtiter plate colorimetric assay, we tested biofilm formation in nutrient-rich (LB) and nutrient-poor (M02) media. Biofilm production is mostly shaped by the genetic background, irrespective of the growth medium (S8D Fig). Yet, in all strains but one, introduction of K1 reduced biofilm formation. Conversely, the replacement of K1 by any other K type increased biofilm formation (Fig 5C and squares in S8D Fig). This suggests that thicker K1 capsules hinder adhesion potentially by masking fimbriae at the cell surface [57,58]. We also noted some strain specificities depending on the K type, notably a strong increase of *Kpn* BJ1 K1 swap's biofilm formation. We posit this could be due to specific physico-chemical properties or alterations between the K1 capsule, other extracellular polysaccharides and matrix compo-nents secreted by BJ1.

A major capsule-associated phenotype is hypermucoviscosity (HMV). Typically linked to the presence of the *rmp* locus [59,60], it has been mostly observed in K1/K2 capsule types and is likely necessary but not sufficient for hypervirulence [61]. Recent work also associates HMV with other K types including K3 [62]. Thus, we tested whether HMV is determined by the capsule type. To statistically compare results across strains we chose a quantitative (slow centrifugation) rather than a qualitative (string test) method. Overall, introduction of the K1 capsule locus tends to increase HMV (Fig 5D), with the exception of CIP 52.145, which remained unaffected. Of note, replacement of the native K1-capsule NTUH K2044 with K1 from SA12 with 3 SNPs in *wzc*, known to impact HMV [63,64], also led to reduction of HMV. Conversely, replace-ment of the native K1 with any other K type in *Kpn* NTUH K2044 reduced HMV (orange squares-Fig 5D, S9 Fig). Finally, introduction of K3 capsule type, not only did not increase HMV, but significantly reduced it, which contrasts with a trend observed in Beckman and colleagues [62]. Sequence analyses of the K3 sequence used in this study (Table 1, K3 ref-erence strain) revealed a truncation of ~10% of the N-terminal region of the CpsACP protein, which could partly explain these discrepancies.

Finally, we analyzed several phenotypes taking into account the presence of *rmp* in the capsule-swapped strains. In our capsule-swapped strains, three genetic backgrounds (NTUH K2044, BJ1 and CIP 52.145) are *rmp*+, while two

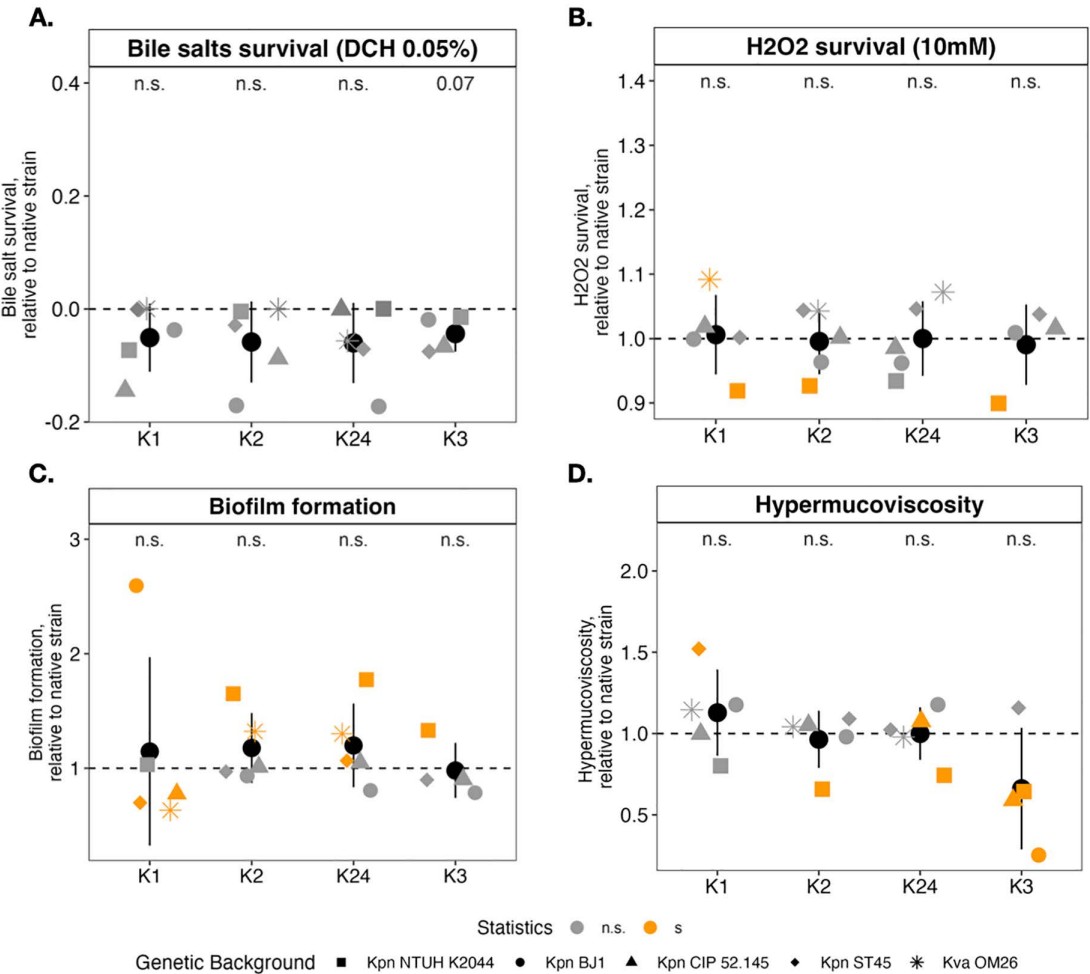

**Fig 5. Virulence-associated traits of capsule-swapped strains.** Survival of capsule-swapped strains to 0.05% deoxycholate (A) and to 10 mM $H_2O_2$ (B), the ability to form biofilm in nutrient-rich media (C), or hypermucoviscosity index (D) relative to their respective native strain. Each point represents the mean of at least three independent biological replicates of each capsule-swapped strain. The color (orange or gray) indicates significant differences or not ($p < 0.05$), relative to their own native strain. Shape of dots correspond to the different native strains (genetic background). The average across K type is indicated by the black dot ($N = 5$). $*p < 0.05$; $**p < 0.01$; $***p < 0.001$, One-sample $t$ test, difference from 1. The data underlying this Figure can be found in S1 Data.

backgrounds are *rmp-* (ST45 and OM26). As expected, swaps with *rmp* produced more capsule (S10A Fig). Whereas no differences were observed in biofilm formation in nutrient-rich medium, strains growing in poor medium and encoding the *rmp* locus, when capsule production is most important, form significantly less biofilm (S10B Fig). This may be due to increased hypermucoviscosity which tends to reduce adherence surfaces and favor inter-cellular interactions. Overall, all *rmp*+ strains are more HMV than all *rmp−*, with one exception, Kpn BJ1 encoding the K3 capsule (S10C Fig). Interestingly introduction of K24, a relatively thin capsule in the two K2-strains, resulted both in increased HMV and in string-positive colonies (S9 Table).

Collectively, our results show that capsule types show a certain degree of conserved properties across different genomic backgrounds, mostly resulting in changes in biofilm formation and hypermucoviscosity. This underscores the

modularity of the capsule, as K type-specific phenotypes are maintained upon transfer and integration in various genetic backgrounds.

## Discussion

Our study investigates the evolution of complex functions and the consequences upon integration into a new host genome. The pervasive exchangeability of capsule types within and across species is mostly driven by biotic stresses, host immunity, phage predation, and nutrient availability [25,31,65]. Such exchangeability implies an underlying modularity of the genome architecture, allowing fast adaptation to changing conditions. Indeed, for capsule swaps to be advantageous, these exchanges should result in minimal fitness costs, whilst conserving their functionality (and thus, expression) across different genetic backgrounds. Here, we undertook an integrative approach and analyzed a collection of capsule-swapped strains to present evidence for a clear example of plug-and-play evolutionary dynamics in Bacteria.

First, despite the tight link between the capsule and the central carbon metabolism [39,50], no regulatory network rewiring of the latter was observed. Transcriptomic analyses revealed minimal gene expression changes upon introduction of a novel K type. The expression of the capsule locus itself does not change across the different genetic backgrounds but is mostly specific to each K type (PCA-S5 Fig). Accordingly, capsule production followed a conserved hierarchy across K types –K1 > K2 > K24 > K3–, regardless of the host genome. K type-dependent differences in capsule production may arise from multiple, non-exclusive mechanisms. Variation in K type-specific gene content, particularly in *wzy* encoding the polysaccharide polymerase, can influence capsule chain length and thickness [66]. Intrinsic properties of the capsular polysaccharide—such as linear versus branched structures—may influence capsule density and packing. In addition, sequence variation in conserved K-locus promoters could change transcription levels, although expression of non-conserved genes could not be directly quantified. A similar conserved hierarchy existed across genetic backgrounds, regardless of K types. Indeed, the three genetic backgrounds associated with the highest capsule production, are hypervirulent strains (NTUH K2044, BJ1, and CIP 52.145). This suggests a very tight regulation of capsule genes, dictated both by the capsule type and by the genetic background, but with limited, or no interaction.

Second, both growth measurements across different environments and direct competitions show no significant differences between capsule-swapped strains compared to their native counterparts. These findings align with prior work showing that most HGT events have only minor fitness effects under laboratory conditions when integrated at a neutral position [67]. This is also in line with the evolutionary dynamics observed during the first steps of co-adaptation upon acquisition of a novel K type. Specifically, our evolution experiment shows that the maintenance of a K type depends more on environmental conditions (nutrient availability) than on a shared life-history between the capsule locus and its genetic background. Indeed, we show that all capsulated strains follow similar evolutionary trajectories suggesting that novel capsules are ready-to-use and require no co-adaptation to the genome. Additionally, the molecular mechanisms underlying adaptation to novel environments are similar in both capsule-swapped and their respective native strains, as shown by mutations emerging in parallel across all strains. These mutations result in the modulation of capsule expression, without altering the capsule biochemistry. Our data shows that capsule swaps have surprisingly little impact on the expression of other functions. Furthermore, the K type specificities are maintained across genetic backgrounds. This modularity, relative to the rest of the genome, contributes to explain the high rates of swaps found in natural populations.

Finally, key phenotypes and functions associated to capsule K types should persist across genetic backgrounds. An early study in which capsule-swapped strains were generated by recombining large chromosomal segments encompassing the capsule locus [41] showed that not all virulence-associated traits were inherited with the K type. Yet, this study postulated that the role of capsule in *Kpn* virulence is likely multifactorial and dependent on epistatic interactions with the genetic background [41]. In more recent work, Huang and colleagues, also performed exchanges of capsule loci in *Klebsiella* [35], and specifically tested virulence phenotypes. Their results bring further proof to the plug-and-play mechanism of capsule evolution, as they clearly show that some capsule loci alone can encode specific virulence functionalities. Indeed,

survival profiles within macrophages and during liver infection using capsule-swapped strains revealed that virulence levels—high or low—were dictated by K type alone, irrespective of genetic background [35]. Similarly, we show that the K1 capsule locus alone generically increases HMV and diminishes biofilm formation. Among all other phenotypes tested, we could not observe any other specific K type-function associations. Expanding the panel to include more ecologically diverse K types may uncover other serotype-specific functions. Also, more complex selection pressures encountered *in vivo*, such as competition within complex microbial communities or growth under anaerobic conditions, may reveal novel K type-specific functions. Indeed, we had previously shown that exchange of K type could also result in exchange of susceptibility against phage infection [44]. Hence, capsule swaps very specifically change some bacterial phenotypes without affecting other processes.

We nevertheless detected some context-dependent epistasis. Introduction of the same K1 capsule locus into two K2 genetic backgrounds (BJ1 and CIP 52.145) resulted in contrasting outcomes: minimal transcriptomic changes in *Kpn* BJ1 versus widespread regulatory shifts in *Kpn* CIP 52.145. More striking, despite numerous attempts, we were unable to introduce the K3 capsule type cloned from *Kpn* ATCC13883T into *Kva* OM26 strain suggesting that some capsule swaps may be highly deleterious or even lethal. Such incompatibilities may reflect underlying differences in core metabolic gene content, as each *Klebsiella* taxon harbors a distinct metabolic profile [68]. Indeed, both K2 strains as well as *Kpn* ATCC13883T and *Kva* OM26 belong to different phylogenetic sublineages and subspecies, respectively. These metabolic differences [68] suggest that capsule-host compatibility may depend on specific metabolic pathways that help mitigate negative epistasis. However, both examples of epistasis may be a consequence of the experimental setup. Capsule-swapped strains were constructed through allelic replacement by recombination at conserved flanking regions (*galF* and *ugd*), yet, in natural populations, we observed that recombination tracts often extend well beyond the capsule locus [25]. These extended tracts may result in co-transfer of additional loci, such as the O-antigen biosynthesis enzymes encoded on the *rfb* locus [69] just downstream the capsule locus, the *his* locus [70] or other core genes. Co-transfer could contribute to surface structure compatibility, membrane homeostasis, and metabolic integration. A recent study showed that capsule production can be heterogeneous within clonal populations, a variability shaped by both the capsule locus and the genetic background [71]. While insertion of the K1 locus consistently resulted in non-heterogeneous phenotype across genetic background, other K types did not follow a consistent pattern, indicating that some capsule-encoded traits might be governed by epistasis [71]. Nevertheless, examples supporting this model –where capsule integration is restricted by host genome compatibility—remain rare cases.

The capsule is encoded both by core and accessory genomes. A recent study in *Vibrio parahaemolyticus* shows that epistatic interactions between the core and the accessory genomes are rare [72]. The authors indicate that frequently transferred genetic elements evolved in a 'plug-and-play'-like architecture. Indeed, our findings support a plug-and-play model of capsule evolution, in which capsule loci act as modular transferable elements. However, a following study by the authors indicates that *V. parahaemolyticus* capsule locus could be an exception [73]. This is in line with studies on capsule evolution in *Spn*, where fitness outcomes of capsule-switched isogenic strains are often shaped by strong K type-genotype epistasis [74]. These differences between *Spn* and *V. parahaemolyticus* compared to *Kpn* could be due to contrasting evolutionary mechanisms for capsule variation. The former two species are naturally competent, which somewhat limits transfer size and could account for the capsules evolving primarily by intra-locus recombination, generating mosaic or chimeric loci rather than full-locus novelty [74]. Although some modularity exists—for instance, flippases in *Spn* show partial interchangeability—this is constrained by substrate specificity [75]. Relaxed specificity can lead to growth defects after recombination. These are likely counter-selected in nature [76], limiting the spread and persistence of deleterious chimeras. Finally, the narrow ecological niche of *Spn* and *V. parahamolyticus* may result in much stronger selection than the more environmentally versatile *Kpn*. Hence, the mechanism of capsule change, i.e., intra-locus recombination, appears to impose greater evolutionary constraints than the whole-locus replacement observed in *Kpn*.

Taken together, the study of a complete macromolecular system rather than isolated gene interactions allowed us to provide an explanation for the pervasive transferability of capsule loci not only across distant *Klebsiella* lineages but also between species and genera. Collectively, our data show that capsule loci can be horizontally transferred and successfully integrated into diverse genetic backgrounds with minimal disruption to cellular fitness or regulatory networks. Furthermore, capsule loci act as modular elements with inherent functional properties which are inherited across genetic backgrounds. Their large diversity and rapid evolution do not seem constrained by epistasis, but instead integrate and persist across diverse genetic backgrounds, exemplifying a true plug-and-play model of evolution.

## Materials and methods

### Bacterial strains and growth conditions

*Klebsiella spp*. strains were grown at 37 °C in 4 mL Luria Bertani Broth (LB Miller)—unless indicated otherwise—in 14 mL tubes and under orbital shaking conditions (250 rpm) or agar plates. Nutrient-poor medium corresponds to minimal medium M63B1 supplemented with 0.2% of glucose (M02). The strains used in this study, as well as their genomic annotations and references, are described in S6 Table.

### Scarless serotype swap

The scarless K type swaps were performed as described before [44].

**Capsule deletion mutant.**  Capsule deletion was performed by recombination at the homologous regions (5′-*galF* and *ugd*-3′, see S7 Table, *Primers*). The different genetic backgrounds, referred to as native strains in the following work, were transformed by electroporation with a λ-red-carrying plasmid (pKOBEG199, see S7 Table, *Plasmids*). Transformants were selected on LB plates supplemented with tetracycline and 0.2% of glucose. Competent cells of transformants were induced with 0.2% L-arabinose for 2 hours, to increase recombination upon transformation by electroporation of the deletion cassette leading to loss of the capsule locus and replacement with the deletion cassette—containing a kanamycin resistance gene, one I-Scel cut site and two FRT sites—and selected on LB with kanamycin at 37 °C. Noncapsulated colonies were then selected and transformed by electroporation with pMPIII, a plasmid encoding a flippase (FLP) (see S7 Table, *Plasmids*). The kanamycin resistance gene was excised by the FLP flippase acting on the FRT sites leaving a small scar containing a I-Scel cute site for the swap step. Cells were plated on LB supplemented with spectinomycin and incubated at 30 °C. A colony was then grown overnight at 42 °C to cure the pMPIII plasmid then plated on LB. Noncapsulated clones that grew only on LB were selected.

**Generating pKAPTURE vectors.**  Capsule cloning was done using a linear cassette named pKAPTURE consisting of two homolog regions in reverse (5′-*galF* and *ugd*-3′), an origin of replication, a kanamycin resistance gene and two I-Scel cut site. The cassette circularizes around the capsule locus *via* recombination and captures the whole-locus to form a circularized pKAPTURE. To do so, the linear cassette was transformed in electrocompetent pKOBEG199-strain, in which expression of l-red had been induced (see above). Cells were then plated on LB with kanamycin and incubated at 37 °C. Kanamycin-resistant transformants are expected to carry recircularized pKAPTURE containing the capsule of the native strain. These transformants were then grown overnight supplemented with kanamycin and EDTA (to limit capsule expression). pKAPTURE plasmids were extracted and electroporated into the abovementioned capsule deletion mutant to generate a stock of pKAPTURE expressing a given capsule type. After recovery, cells were plated on LB supplemented with kanamycin and allowed to grow overnight at 37 °C. Capsulated colonies were restreaked in parallel on LB and LB supplemented with kanamycin (50 μg/mL). Capsulated colonies on LB supplemented with kanamycin but noncapsulated on LB were selected and considered a source of pKAPTURE vector which carries the capsule locus of interest.

**Generating capsule-swapped strains.**  Electrocompetent capsule deletion mutants were transformed with pKAPTURE encoding a specific capsule locus. After recovery, cells were plated on LB with kanamycin and grown

overnight at 37 °C. To allow integration of pKAPTURE, electrocompetent cells of the strain with the pKAPTURE were transformed with pTKRED (plasmid encoding RecA and I-SceI enzymes, see S7 Table, *Plasmids*). Cells were recovered at 30 °C in LB supplemented with kanamycin, to avoid pKAPTURE loss, and 0.2% glucose for 1 hour 30 min, prior to plating on LB with kanamycin, spectinomycin and 0.2% glucose and incubated at 30 °C. To induce integration, individual colonies were resuspended in M63B1 supplemented with spectinomycin, 0.2% L-arabinose, 0.2% glycerol and grown at 30 °C. I-SceI endonuclease cuts the chromosome and the circularized pKAPTURE containing the capsule locus at the I-SceI sites. *recA* expression increases recombination leading to chromosomal repair by of the capsule locus carried by the pKAPTURE. After 12–24 hours, cells were diluted depending on culture turbidity, plated on LB supplemented with 0.2% glucose and grown at 42 °C to cure pTKRED. Capsulated colonies identified were restreaked in parallel on i) LB, ii) LB with kanamycin, and iii) LB with spectinomycin. Capsulated clones that only grow on LB are considered successful swapped clones. All swapped clones were verified by Illumina sequencing and their sequences were compared to native strains using *breseq* v.0.35.7 [77] with default parameters (S8 Table). Strains in which the original K type was introduced in the acapsulated mutant were also generated as controls and are referred to as complemented strains.

## Capsule loci genetic relatedness

Only genes conserved across all the five K types (*galF*, *cpsACP*, *wzi*, *wza*, *wzb*, *wzc*, *wcaJ*, *gnd*, *manC*, *manB*, *ugd*) were included in the analysis. Protein sequences of homologous genes were compared pairwise in both directions (e.g., *galF*_K1 aligned on *galF*_K2 and *galF*_K2 aligned on *galF*_K1) using BLAST (v.2.16.0+). The mean percentage identity was calculated for each pairwise gene comparison. Finally, the overall capsule locus pairwise percentage identity was determined.

## Capsule extraction and quantification

To measure the total amount of CPS produced, we followed the protocol described before [78] and quantified by the uronic acid method [79]. Briefly, $OD_{600}$ of overnight cultures in LB medium was measured. Then, 500 µL were transferred to an Eppendorf tube with 100 µL of Zwittergent 1% in 100 mM citric acid and placed in 56 °C dry bath for 20 min. After centrifugation, 300 µL of the supernatant was transferred to a new tube. To precipitate the polysaccharides, 1,200 µL of cold ethanol was added, kept at 4 °C for 20 min and centrifuged at high speed (14,000 rpm). The pellet was washed with 70% ethanol and allowed to dry at 56 °C, resuspended in double-distilled water, and incubated in a 56 °C dry bath to facilitate dissolution (as described in Domenico and colleagues [78]). The uronic acid concentration of each sample was determined from a standard curve of glucuronic acid. Briefly, 1,200 µL of 0.0125 M sodium tetraborate dissolved in $H_2SO_4$ was added to 200 µL of samples and uronic acid standards, and placed 5 min in boiling water. After cooling down on ice, 20 µL of 0.15% 3-phenylphenol dissolved in 0.5% NaOH was added to the different samples. 200 µL from each tube was transferred to a 96-well microtiter plate and the absorbance was read at 520 nm. Finally, capsule production was normalized to the culture $OD_{600}$ to account for differences in cell density. Of note, we have previously shown that uronic acid measurements for the capsule K types studied here, perfectly correlate with total cell volume and capsule thickness [44].

## Capsule area

**India ink staining.** 10 µL of overnight culture was mixed to 5 µL of India ink on a microscope slide. The coverslip was carefully placed on the dye-bacteria mixture. Using a paper towel, pressure was carefully applied to press down on the coverslip until the slide appeared as a photographic negative.

**Visualization and measurement.** Cell morphologies were imaged using a Zeiss Axio Imager.M2 microscope, with ×100 magnification, equipped with an Axiocam 503 mono camera (Carl Zeiss, Germany). Images were acquired using the ZEN lite software (v. 3.6). ImageJ software (v. 2.16.0) was used to measure the capsule area (N ~ 100)

 

by adjusting contrast and using the *Analyze particles* function. Due to methodological limitations, we were not able to properly measure the capsule area of strain ST45_K24 and ST45_K3 which appeared too thin and not reliably identified by the software. For similar reasons, few events could be counted for NTUH K2044_K24 and NTUH K2044_K3.

## Growth curves

Overnight cultures were diluted at 1:100 in the different growth environments. 200 μL of each subculture was transferred in a 96-well microtiter plate and allowed to grow at 37 °C, under orbital shaking for 16 hours. Absorbance ($OD_{600}$) of cell cultures was measured every 15 min with a TECAN Genios plate reader.

## HMV

HMV was tested first by the string test (S9 Table), but was not pursued further due to its qualitative nature and limited reproducibility [39, 60]. We therefore selected the sedimentation assay as the primary measure of HMV, as it provides an objective, quantitative, and reproducible assessment [39].

**Sedimentation assay.** Precultures of each strain were done by inoculating fresh LB medium with a colony, grown overday and then diluted to 1:200 in M02 for overnight culture. To initiate the experiment, cultures were vigorously vortexed and 200 μL of culture was transferred into a 96-well microtiter plate. Absorbance ($OD_{600}$) was measured to set the initial OD ($OD_i$). Cultures were then sedimented by slow centrifugation (1,260 x $g$) for 5 min at room temperature. 200 μL of the top part of culture was then transferred into a 96-well microtiter plate. Absorbance ($OD_{600}$) was measured to set the final OD ($OD_f$). Background noise ($OD_{600}$ of fresh medium) was subtracted from each measurement, and the hypermucoviscosity index was calculated as ($OD_f/OD_i$).

## Biofilm formation staining using crystal violet

The study of biofilm formation ability was performed as described before [80]. Strains were grown overnight in LB and diluted at 1:100 in the different growth environments. 200 μL of each subculture was transferred in a 96-well microtiter plate then incubated at 37 °C for 24 hours. Supernatant was removed by flicking the plate and bacteria were washed thrice with water. Extra water was emptied by flicking. The surface-attached bacterial mass remaining in the well was stained with 220 μL of crystal violet (1%) for 30 min at room temperature, washed thrice with water and air-dried. The bacterial mass was resuspended in 220 μL of ethanol:acetone (80:20) and absorbance ($OD_{590}$) was measured.

## Bile salts survival assays

Sensitivity to bile salts was measured as previously described [81]. Briefly, overnight cultures in LB were serially diluted and plated on LB or on LB supplemented with either 0.05% (physiological conditions) or 0.5% of CHO (primary bile salts, sodium cholate) or DCH (secondary bile salts, sodium deoxycholate). Colonies were allowed to grow at 37 °C and surviving CFU were counted after 24 hours.

## Hydroxide peroxide survival assay

To assess survival to oxidative stress [82], overnight cultures in LB were diluted at 1:100 in LB and subcultures were grown overday at 37 °C until OD ~0.6–0.8. 500 μL of overday culture was transferred in a 96-deep well plate supplemented with either 0, 5, 10, or 15 mM of $H_2O_2$. Cultures were incubated for 1 hour at 37 °C without shaking. Subcultures were serially diluted, spotted on LB agar plates and grown overnight at 37 °C. Surviving CFU were counted after 24 hours and compared to the control without $H_2O_2$.

## Evolution experiment

Three independent clones of each strain were used to initiate each of the three evolving populations in LB and nutrient-poor (M02) media. Each day, populations were diluted 1:100 into fresh media and allowed to grow for 24 hours at 37 °C. In parallel, cultures were serially diluted, plated and visually inspected each day on LB to count capsulated and acapsulated clones. The experiment was performed for 15 days accounting for an estimated 100 generations (= 15 days × 6.7 generations/day) (based on *Escherichia coli* generations/day). Although each growth medium has slightly different carrying capacities, all cultures reached bacterial saturation before daily passaging, ensuring that the different populations underwent a similar number of generations across growth media.

## Fluorescent strains construction

Scarless chromosomal integration of the *mCherry* fluorescent reporter was performed by double recombination event at the neutral attTn7 site located downstream of the *glmS* locus [83]. Briefly, ~500 bp regions flanking the attTn7 site downstream of *glmS* were amplified by PCR, along with the *mCherry* gene under the control of the strong constitutive pLpp promoter (See S7 Table, *Primers*). To assemble the fragments into a vector (pKNG101), we used GeneArt Gibson Assembly HiFi kit (Invitrogen) and incubated the mix for 30 min at 50 °C. The assembly reaction was diluted 1:4 and electroporated into competent *E. coli* DH5α strain and selected on LB plates with streptomycin (100 µg/mL). Colonies with integrated fragments were checked by PCR, extracted, electroporated into *E. coli* MFD λ-pir strain, and used as a donor strain for conjugation in capsule-swapped strains. Single cross-over mutants (transconjugants) were selected on streptomycin plates (200 µg/mL) and double cross-over mutants were selected on LB without salt and supplemented with 5% sucrose after growth at room temperature. Mutants were verified for their sensitivity to streptomycin and by PCR, and *mCherry* expression confirmed by fluorescent microscopy.

## Bacterial competitions

Precultures of each strain were done by inoculating fresh LB medium with a colony. They were grown overday in LB and then diluted to 1:100 in M02. Strains with or without fluorescent tags were mixed in a 1:1 ratio. The mixes were diluted to the 1:100 either into M02 or cold PBS 1X to avoid further growth and verify cell ratio in the competition mix using flow cytometry ($T_0$). The M02 plate was placed at 37 °C in shaking conditions in a microtiter plate reader to allow bacterial growth and competition. After 24 hours ($T_{24}$), samples were diluted 1;100 into cold PBS 1X, and the proportion of each strain was assessed by fluorescence-activated cell sorting (FACS) analysis using CytoFlex S. Four competitions between each genotype were performed, two in which one of the strains was tagged *mCherry*, and the other two with the *mCherry* integrated in the other competitor.

## FlowJo analyses and fitness calculation

Flow cytometry data were analyzed using FlowJo software (v10.10.0). Subsequent analyses were performed using R.4.4.1. The relative fitness was calculated by dividing the ratio of cells at $T_{24}$ compared to $T_0$. Competitions with a relative fitness <0.5 and >2, most likely indicative of a technical mistake, were removed. These accounted for 35 out of 346 competitions, evenly distributed across competitions. Control experiments to assess whether *mCherry* integration was associated to a cost revealed no difference in fitness (S6C Fig).

## RNA extraction and sequencing

**Extraction.** An overday culture was started in M02 from freshly plated colonies and incubated at 37 °C until $OD_{600}$ reached ~0.6–0.8. RNA was extracted using the Macherey Nagel Trizol (ref:740971.250) according to the manufacturer's instructions and treated with DNase I provided in the kit. RNA concentration, quality, and integrity from four independent

replicates were checked using the Invitrogen Qubit and the Agilent 2100 Bioanalyzer system. Four independent samples per population were sequenced.

**Sequencing.** Library preparation and sequencing were carried out at the Biomics Platform, Institut Pasteur. cDNA libraries were prepared from 1 to 3 µg of total RNA using the Illumina Total RNA Library Preparation Kit (Illumina, USA), following the manufacturer's protocol. To facilitate rRNA depletion, Ribo-Zero Plus Microbiome probes (Illumina) were used. Index barcodes were added by PCR for 13 cycles. Unbound adaptors and index primers were removed via purification with AMPure XP magnetic beads (Beckman Coulter, USA). The final libraries displayed an electrophoretic size distribution ranging from 250 to 900 bp, with a predominant peak at ~400 bp, as assessed on a 3,500 Fragment Analyzer (Agilent Technologies, USA). Sequencing was performed on a NextSeq 2000 system using a P3 50-cycle flow cell (Illumina) to generate 67-nt single-end, dual-indexed reads.

### RNA sequencing analysis

**Cleaning.** Single-end strand-specific 65 bp reads were cleaned of adapter sequences and low-quality sequences using cutadapt version 4.9 [84] with options "-m 25 -q 30 -O 6 --trim-n --max-n 1". Gene expression quantification was performed using salmon version 1.9.0 with the "-l A" option [85]. For each condition, a specific reference transcriptome was built concatenating both the fasta transcriptomes of the genetic background and the capsular region. Seven out of 96 samples were excluded due to poor sequencing quality, and all strains were represented by at least three biological replicates (raw data accession number: GSE306874 in https://www.ncbi.nlm.nih.gov/geo/ repository).

**Gene expression analysis.** Gene expression data were analyzed using R version 4.3.2 [86] and the Bioconductor package DESeq2 version 1.42.1 [87]. The data structure was explored using a Principal Component Analysis based on the replicate-adjusted variance-stabilized transformed count matrix. Replicate adjustment was performed using the removeBatchEffect() function from the limma R package [88]. The normalization and dispersion estimation were performed using the default parameters and statistical tests for differential expression were performed applying the independent filtering algorithm. Four background-specific independent analyses were conducted to compare the gene expression across conditions, i.e., between the different capsule-swapped strains within a given genetic background. For each background strain, a generalized linear model, including the replicate effect as blocking factor, was set to test for the capsular swap effect on gene expression. For each pairwise comparison, shrinkage of the $\log_2$(FC) was performed using the ashr method [89], raw p-values were adjusted for multiple testing according to the Benjamini and Hochberg procedure [90] and genes with an adjusted p-value lower than 0.05 were considered differentially expressed.

Several quality controls were performed. First, PCA on the expression of core capsule genes revealed that the first component could differentiate acapsulated strains from all other capsulated strains with high inertia (S5A Fig). Secondly, in Δcap strains, all capsule genes were found highly downregulated due to their absence. Thirdly, capsule genes which are specific to each of the native capsule locus and absent in the swapped strains were significantly downregulated. Conversely, upregulated capsule genes were mostly genes specific of the newly acquired capsule locus and not found in the native one. Finally, PCA using only the core capsule genes allowed the clustering of independent biological replicates together S5B Fig.

### Whole genome sequencing

The genomes of capsule-swapped strains and clones producing reduced capsule amounts were extracted using the guanidium thiocyanate method [91] prior to Illumina sequencing. Their sequences were compared with ancestral genotypes using *breseq* v.0.35.7 [77] with default parameters. For clones producing reduced capsule amounts, some mutations *(tatC, wcaJ)* were further verified by PCR (See S7 Table, *Primers*) and subsequent Sanger sequencing. Full list of identified mutations is provided in S5 Table (raw data accession number: PRJNA1365496 in https://www.ncbi.nlm.nih.gov/bioproject/ repository).

## Other software and packages

All the data analyses were performed with R version 4.4 and Rstudio v2022.02.1, except when precised otherwise in Methods. For data frame manipulations, we used dplyr v1.1.4 along with the tidyverse packages v2.0.0. We used the packages ggpmisc v.0.6.0 and Kendall v2.2.1 for the linear regressions and fitness transitivity, respectively. Graphs were performed using ggplot2 v.3.5.1, gridExtra v.2.3, and ggtext v.0.1.2.

## Supporting information

**S1 Fig. Genomic organization of capsule loci of strains used in the study.** Capsule locus identification and annotation were done using Kaptive [36]. Alignment and visualization were done using Clinker [92] (https://github.com/gamcil/clinker) with modifications. Small gray arrows indicate promoters of the locus. Large gray arrows represent nonannotated proteins with no homologs in the other capsule loci, whereas arrows in shades of dark blue correspond to hypothetical proteins with homologs in other strains. K types labeled as 'Alternative loci', corresponding to *Kpn* NTUH K2044 and *Kpn* CIP 52.145, are native loci to their strains but not used as template to generate the capsule-swapped strains. (DOCX)

**S2 Fig. Capsule production of capsule-swapped strains determined by the uronic acid method.** A. Comparison of capsule production between the wild type strain and its complemented strain corresponding to the reintroduction of either the native capsule locus (green) or an alternative capsule locus of the same K type (pink). Values were normalized by an $OD_{600}$ of 1. *P*-values correspond to unpaired Wilcoxon test. B and C. Capsule production across different capsule loci (B) or genetic background (C), ranked from lowest to highest capsule production. Blue lines represent regressions for each K type ($R^2$), and the surrounding gray area indicate the standard error. Each dot corresponds to an independent biological replicate. The data underlying this Figure can be found in S2 Data. (DOCX)

**S3 Fig. Correlation between capsule thickness and capsule quantification.** A. Representative images of *Klebsiella* strains under the microscope. India ink staining images of Kpn CIP 52.145, Kpn NTUH K2044, and Kpn ST45 at a magnification of 100× after overnight culture in LB. Scale bar = 10 μm. B. Capsule area positively correlates with capsule production. Each panel represents each set of capsule-swapped from the three independent genetic backgrounds measured. Capsule thickness was determined by measuring the exclusion area generated by the capsule. This area was quantified from microscopic images of India ink-stained cells using ImageJ. The lightly capsulated cells could not be measured confidently and are thus not included in the capsule thickness analyses. The size of points indicates the number of cells analyzed (between 50 and 100, except for the very lightly capsulated strains). Capsule production was estimated by glucuronic acid method from capsule extracts (see Materials and methods). C. Correlation of all capsule-swapped strains together, irrespective of their genetic background. Statistical analyses for panels B and C were performed using a linear model (LM) implemented with the *smooth* function in R. The data underlying this Figure can be found in S2 Data. (DOCX)

**S4 Fig. Transcriptomic analyses of capsule loci genes in capsule-swapped strains.** $Log_2FC$ (fold change) of capsule locus genes, i.e., genes present in any of the five capsule loci types considered. The shape indicates the significance of the adjusted *p*-value and the color represents the capsule types. The genetic background and the native capsule types are indicated at the top of each subpanel. Capsule genes indicated on the x-axis are colored considering their K type specificity: dark red for K1-specific genes, *wzy* is indicated in brown (identified for K1, K24, and K64), orange for K2, light blue for K24, red for K3, *wbaZ* is indicated in purple (identified in K3, K24, and K64 serotypes) and dark blue for K64-specific genes. Core capsule genes (present in all K types) are in black. The data underlying this Figure can be found in S2 Data. (DOCX)

**S5 Fig. Adjusted Principal Component Analysis (PCA) of gene expression.** Components 1 and 2 are displayed and are followed by the percentage of variance explained by each component. Each dot represents a biological replicate annotated as follows: replicate number (EX), underscore and K type (KX). PCA analysis was performed on the core capsule genes (A and B) and the core genome (C). For better visualization of the clustering of capsule-swapped strains, the PCA on the core capsule was done with (A) and without (B) taking into account the dCap (acapsulated) strains. The data underlying this Figure can be found in S2 Data.
(DOCX)

**S6 Fig. Growth and fitness of capsule-swapped strains.** A and B. Growth curves (error bars are not included for visibility reasons) (A) and the area under the growth curve (AUC) (B) of capsule-swapped strains relative to their respective native strain (dotted line) in either nutrient-rich (LB) or nutrient-poor (M02) media. Gray dots represent individual biological replicates ($N=5$). C. Pairwise competitions in nutrient-poor medium (M02) of fluorescent versus nonfluorescent strain (one-sample $t$ test, difference from 1). Asterisk beside K loci (*) indicates the native serotype of each strain. Δcap indicate acapsulated control strains. The data underlying this Figure can be found in S2 Data.
(DOCX)

**S7 Fig. Evolutionary fate of the capsule in capsule-swapped and native strains.** A. Proportion of capsulated clones throughout the 15 days of evolution of parental strains and their respective isogenic capsule-swapped strains before daily passages of each culture either in nutrient-rich (LB, green line) or nutrient-poor (M02, blue line) media. Bold lines represent the average of the independent populations of the same strain grown in a given environment. Gray lines represent each of the independent populations. B. Proportion of capsulated clones throughout the first 5 days of evolution in nutrient-rich medium. Each line represents the average of at least three independently-evolving populations. The data underlying this Figure can be found in S2 Data.
(DOCX)

**S8 Fig. Survival to biotic stress of capsule-swapped strains.** A and B. Capsule-swapped strain's survival to 0.5% cholate (CHO) or deoxycholate (DCO) (A) and to 5 mM or 10 mM $H_2O_2$ (B), relative to their respective untreated condition. *$p<0.05$; **$p<0.01$; ***$p<0.001$, one-sample $t$ test, difference from 100. C. Native capsulated strains and their respective acapsular mutant (Δcap) survival to 5 or 10 mM $H_2O_2$, relative to their respective nontreated condition. ns, nonsignificant, two-sample paired $t$ test. D. Biofilm formation of capsule-swapped strains in nutrient-rich (LB) or nutrient-poor (M02) media, relative to their respective native strain. Shape of dots correspond to the genetic background; the K type is indicated on the x-axis and identified by the color. Each point represents the mean of at least three independent biological replicates. ns: nonsignificant; one-sample $t$ test, difference from 1. The data underlying this Figure can be found in S2 Data.
(DOCX)

**S9 Fig. Hypermucoviscosity index of capsule-swapped strains.** A. Raw values of the hypermucovsiscosity index (HMV) of capsule-swapped strains, as measured after growth in nutrient-poor medium. B. Hypermucovsiscosity index of capsule-swapped strains relative to their respective native strain in nutrient-poor medium. The serotype is indicated on the x-axis and by the color. Gray points represent independent biological replicates and color points represent the mean of these biological replicates. Asterisk beside K loci (*) indicates the native serotype of each strain. ns: nonsignificant; one-sample $t$ test, difference from 1. The data underlying this Figure can be found in S2 Data.
(DOCX)

**S10 Fig. Experiments analyzed considering the presence (*rmp+*) or absence (*rmp−*) of the *rmp* locus in the genetic background of each strain.** A. Capsule production measured by the uronic acid method and normalized by the $OD_{600}$. B. Biofilm formation, quantified by crystal violet, after 24 hours growth in either nutrient-rich (LB) or nutrient-poor

(M02) media. C. Hypermucoviscosity index upon slow centrifugation. The data underlying this Figure can be found in S2 Data.
(DOCX)

**S1 Table. Statistical analyses – multifactorial ANOVA for all traits analyzed.**
(XLSX)

**S2 Table. Statistical analyses – stepwise regression for all traits analyzed.**
(XLSX)

**S3 Table. Differentially expressed genes (DGE).** List of all DGE in capsule-swapped strains compared to their respective native strains considering |log2 fold-change| > 1.
(XLSX)

**S4 Table. Mutations identified in intermediately capsulated clones from evolving populations during adaptation to nutrient-rich medium.**
(XLSX)

**S5 Table. Sequencing analysis of evolved clones.** Evolved clones sequences were compared to their respective parental strains using *breseq* v.0.35.7 [77].
(XLSX)

**S6 Table. List of strains used or constructed for this work.**
(XLSX)

**S7 Table. List of primers and plasmids used to perform scarless serotype swap and fluorescent strains construction.**
(XLSX)

**S8 Table. Sequencing analysis of capsule-swapped strains.** Capsule-swapped strains sequences were compared to their respective native strains using *breseq* v.0.35.7 [77].
(XLSX)

**S9 Table. Hypermucoviscosity measured by a string test.**
(XLSX)

**S1 Data. Data underlying the main figures.**
(XLSX)

**S2 Data. Data underlying the supplementary figures.**
(XLSX)

## Acknowledgments

We are grateful to Amandine Nucci for technical aide with RNA extractions. We also thank Y. Vitrenko, L. Lemée, and E. Kornobis of the Biomics Platform, C2RT, Institut Pasteur, Paris, France, supported by France Génomique (ANR-10-INBS-09) and IBISA for the samples QC, libraries and sequencing. We thank P.H. Commère, S. Schmutz, and S. Novault of the Institut Pasteur Flow Cytometry Platform for the training, help and guidance. We thank Fabienne Benz for the gift of the *mCherry* gene, and Jean-Marc Ghigo and Christophe Beloin for providing lab space and support. We thank Simonetta Gribaldo for the use of the Zeiss Axio Imager.M2 microscope.

We thank Benoit Pons, Matthieu Haudiquet and Samay Pande for critical reading of the manuscript and Basile Beaud Benyahia and Cyril Anjou for scientific discussion.

The artificial intelligence ChatGPT was used to improve writing style and grammar of the manuscript.

## Author contributions

**Conceptualization:** Julie Le Bris, Olaya Rendueles.

**Data curation:** Julie Le Bris, Olaya Rendueles.

**Formal analysis:** Julie Le Bris, Hugo Varet.

**Funding acquisition:** Olaya Rendueles.

**Investigation:** Julie Le Bris, Olaya Rendueles.

**Methodology:** Julie Le Bris, Hugo Varet, Olaya Rendueles.

**Project administration:** Olaya Rendueles.

**Resources:** Julie Le Bris, Eduardo P.C. Rocha, Olaya Rendueles.

**Software:** Julie Le Bris, Hugo Varet.

**Supervision:** Olaya Rendueles.

**Validation:** Julie Le Bris, Olaya Rendueles.

**Visualization:** Julie Le Bris, Hugo Varet.

**Writing – original draft:** Julie Le Bris, Olaya Rendueles.

**Writing – review & editing:** Julie Le Bris, Hugo Varet, Eduardo P.C. Rocha, Olaya Rendueles.

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
