## [Editor Report · Decision Letter 0]

17 Nov 2025

Dear Dr Le Bris,

Thank you for submitting your manuscript entitled "Serotype swapping in Klebsiella spp. by plug-and-play" for consideration as a Research Article by PLOS Biology.

Your manuscript has now been evaluated by the PLOS Biology editorial staff as well as by an academic editor with relevant expertise and I am writing to let you know that we would like to send your submission out for external peer review.

Once your full submission is complete, your paper will undergo a series of checks in preparation for peer review. After your manuscript has passed the checks it will be sent out for review. To provide the metadata for your submission, please Login to Editorial Manager (https://www.editorialmanager.com/pbiology) within two working days, i.e. by Nov 19 2025 11:59PM.

Kind regards,

Melissa

Melissa Vazquez Hernandez, Ph.D.

Associate Editor

PLOS Biology

---

## [Decision Letter · Decision Letter 1]

18 Dec 2025

Dear Dr Le Bris,

Thank you for your patience while your manuscript "Serotype swapping in Klebsiella spp. by plug-and-play" was peer-reviewed at PLOS Biology. It has now been evaluated by the PLOS Biology editors, an Academic Editor with relevant expertise, and by three independent reviewers.

In light of the reviews, which you will find at the end of this email, we would like to invite you to revise the work to thoroughly address the reviewers' reports. As you will see below, all reviewers are positive about the relevance and novelty of the study, yet some concerns have raised during revision. Reviewer 1 is positive about the conceptual advance but raises important concerns requesting for additional validation experiments, like direct morphological evidence of capsule thickness, more complete functional testing of hypermucoviscosity determinants, and consideration of in vivo host models to substantiate evolutionary and virulence relevance. Reviewer 1 also highlights internal inconsistencies in data interpretation that require clarification rather than new experiments. Reviewer 2 views the conclusions as convincing, but gives some suggestions, such as reporting genome sequencing results of engineered strains, exploring potential causes of K1 variability, and adding contextual discussion or simple follow-up assays to strengthen interpretation. Reviewer 3 is likewise positive but requests clarifications or corrective experiments, including revisiting capsule quantification and hypermucoviscosity assay protocols, improving data presentation, and better justifying or expanding analysis of unexpected transcriptomic and phenotypic outliers, while not necessarily calling for extensive new in vivo experimentation. We agree with all reviewer concerns and would require some additional experimental revisions to address them, as we consider that this would strengthen the work.

Given the extent of revision needed, we cannot make a decision about publication until we have seen the revised manuscript and your response to the reviewers' comments. Your revised manuscript is likely to be sent for further evaluation by all or a subset of the reviewers.

**IMPORTANT - SUBMITTING YOUR REVISION**

*Re-submission Checklist*

*Published Peer Review*

*PLOS Data Policy*

*Blot and Gel Data Policy*

Sincerely,

Melissa

Melissa Vazquez Hernandez, Ph.D.

Associate Editor

PLOS Biology

REVIEWERS' COMMENTS

Reviewer #1:

The work systematically investigates the functional and fitness consequences of engineered capsule swaps across diverse genetic backgrounds, combining transcriptomics, short-term evolution, and phenotypic assays. The main finding is that capsule loci can function as modular, "plug-and-play" units with minimal disruption to the host, which provides a plausible explanation for their widespread exchangeability in nature. The study further reports that key virulence traits associated with specific serotypes are conserved after transfer, and that adaptation to capsule cost follows general characteristics rather than serotype specificity. The manuscript is logically organized, and its findings provide important insights into the evolutionary consequences of capsule exchange. However, all evidence is derived from in vitro experiments. The consequences of capsule swapping in an in vivo host environment remain unaddressed. There are several issues in the manuscript.

Main points:

1. Direct evidence (e.g., by electron microscopy or capsule staining) is needed to support the claim that higher capsule production in K1/K2 corresponds to greater physical thickness, as the manuscript consistently equates production levels with thickness without morphological or quantitative validation. Additionally, the discussion should address the molecular basis for serotype-dependent production.

2. Are Figure 1B and 1C generated from the same dataset and processed differently? If so, why does Figure 1B show the highest K2 capsule production in the Kpn CIP 52.145 background, while Figure 1C indicates the highest production for K1 in the same background?

3. The results in Figure 2 indicate that the capsule gene expression is fairly constant independently of the genomic context. However, findings in Figure 1 show that capsule production is genetic background-dependent. This appears somewhat contradictory. How can the stability of capsule gene expression across different backgrounds be reconciled with the significant differences in capsule production?

4. Regarding Figure S6B, the native Kpn BJ1 strain is K2 serotype. Similarly, there is a capsule-swapped strain in the Kpn BJ1 background that also carries the K2 locus. However, in the short-term evolution experiment, the swapped K2 strain showed a significant increase in acapsular clones. What might explain this observation?

5. In the investigation of the hypermucoviscosity phenotype conferred by different capsule types across genetic backgrounds with or without the rmp locus, was the entire rmp locus systematically analyzed? Figure S8 only presents data for rmpA, yet according to literature, the rmpD gene is also crucial for the mucoid phenotype. Additionally, the string test should be performed to provide direct phenotypic confirmation of hypermucoviscosity.

6. Only the K1 capsule confers a conserved hypermucoviscous phenotype across genetic backgrounds does not support the broad conclusion that "hypermucoviscosity was conserved across genetic backgrounds." Therefore, thise statement appears overstated and lacks the precision.

7. The conclusions are based solely on in vitro evidence, without an assessment of the consequences of capsule swapping in an in vivo host environment.

Minor points:

1. There is no citation for Figure 4B in the manuscript.

2. The presentation of some data lacks the optimal clarity. For instance, representing bacterial growth curves solely by the area under the curve (AUC) metric, rather than showing the raw OD values over time, obscures the actual growth dynamics.

Reviewer #2 (Alvaro San Millan):

In this study, Le Bris and colleagues analyze the phenotypic and transcriptomic effects associated with capsule swapping in different strains of Klebsiella pneumoniae. The capsule locus is commonly exchanged between strains in this species, and the authors hypothesized that capsules may behave as modular, "plug-and-play" units, that produce minimal alterations in the new bacterial hosts while maintaining their functions. In order to test this hypothesis, they constructed a collection of five wild-type trains carrying four different capsule types each, and performed transcriptomic, fitness and evolution assays. Their results convincingly showed that the capsule locus produces minimal effects on the transcriptomic profiles and fitness of the hosts, but conserved serotype-specific traits, such as biofilm formation and hypermucoviscosity. In general, I think this is a timely and relevant study. I particularly appreciate the technical difficulty of constructing the strains, as well as the elegant and robust experiments used. I enjoyed reviewing the paper, and I think it will be of interest for the general readership of PLoS Biology. I provide a few specific comments below:

-Construction of the strain collection. The authors sequenced the genomes of the final strains to ensure that no off-target mutations were acquired during the process, however, they did not present the results. I guess some mutations may have accumulated, and it may be interesting to have a table with those.

-In general, the results with serotype K1 tend to be more variable across different experiments. Could this be due to a rapid rise in capsule-less mutants in rich medium during the experiments?

-Lines 352-354, the authors could easily test the presence of long cells by flow cytometry (using forward scatter values as a proxy) or microscopy.

-lines 388-391, this sentence is confusing.

-Figure 5, any idea why Kpn BJ1 is producing so much more biofilm with the K1 capsule?

-Another interesing example of "plug and play" of a macromolecular system with no phenotypic and transcriptomic effects that could be commented in the discussion can be found here https://academic.oup.com/nar/article/52/20/12565/7816861

Alvaro San Millan

Reviewer #3:

This manuscript by Le Bris, et al. (PBIOLOGY-D-25-03672R1) presents a series of experiments in which they took several Klebsiella strains and introduced 4 different capsule loci into each background to evaluate fitness and other phenotypic consequences of the swapping. An initial observation was that some loci yielded more capsule irrespective of strain background, and that some strains produce more capsule irrespective of capsule type. The authors then examined transcriptomics, growth curves, in vitro competition, rate of capsule loss due to environmental pressures, and phenotypes associated with virulence. The collective observations indicate that swapping capsule loci has very little if any impact on the tested phenotypes. These results suggest that the capsule loci are not broadly restricted to certain strain backgrounds, and "easily integrated" (my phrase) into most strain strains without causing major shifts or the need for adaptation. Previous studies have examined capsule swapped strains but have focused on the role of capsule type in virulence. This manuscript presents a novel angle, trying to understand the evolution of capsule and how horizontal transfer of the capsule locus impacts the recipient strain. This work advances the field of both capsule and Klebsiella biology, suggesting that the number of strain background/capsule type pairs among Klebsiella strains is a consequence of the lack fitness costs associated with interchanging capsule loci. A major strength of this study is in the breadth of strain backgrounds and capsule types examined. Although many of these results feel "negative", the findings are collectively fascinating and important. This manuscript is well-written and well-organized with sound conclusions based on the data presented. A few comments that I hope the authors will find helpful are listed below.

Major Comments

1. Throughout the manuscript, please make axis labels on graphs larger. They are proportionally very small relative to the graph/image size, and in many cases nearly unreadable.

2. One finding I found particularly intriguing is that strain 52.145 with K1 had so many differentially expressed genes, but none of the others had any effect even close to this. I was hoping for more detail/speculation in the discussion, but this was just glossed over. I was not surprised to see that most swaps did not result in much change in expression as the capsule loci do not contain transcriptional regulators, so this result really stood out to me. Also, if I understand the differences between Figs 2B and 2C correctly, it suggests there are~300 CPS-related genes that are differentially expressed? (a) if this interpretation is wrong, then I think the text may need some clarification, (b) if correct, how are CPS-related genes determined? This seems very high to me. Perhaps a bit more detail on which genes were most highly DE would be of interest to other readers!

3. Fig. 2, why was a K1 strain with swapped CPS not included in the differential gene expression analysis? It struck me as odd to include two K2 strains and omit the K1.

4. The paragraph/section beginning on line 305 (and lines 301-303) describes capsule thickness, but thickness was never measured—only amount of capsule. I would encourage limitation of linking amount to thickness.

5. Regarding the biofilm data in Fig 5, I would like to know what the relative biofilm production looks like between the strains (particularly between HMV+ and HMV- strains), not just ratios of the impact of the varying K types. Could the authors provide this in a supplemental figure? A simple bar graph of OD595 would suffice.

6. The sedimentation resistance values for ST45 and OM26 (Fig. S8) are surprisingly high given they are rmp negative. I'm concerned this may have to do with the specific protocol applied (see concern below regarding this), and therefore am somewhat concerned about the interpretation of the data. HMV negative strains should have a post-spin OD600 value below 0.15-0.2. Have the authors verified these strains do not contain wzc mutations that might lead to elevated resistance? Was a capsule mutant included in the assays? This strain should have a post-spin OD600 of nearly zero, so if this strain is much above 0.1, I would revisit the protocol parameters.

7. Although I generally thought this manuscript was well-written, there are numerous typos and a few issues with grammar that I found led to some confusion on my part. I am already late submitting this review, so I'm not detailing them, but I do think it might be helpful to have a fresh set of eyes give it a read through!

I have several concerns about information in the Materials & Methods:

8. I noted a statement that raised two concerns. Lines 594-595 states that a culture was mixed with zwittergent, then the supernatant was discarded to quantify only CPS attached to the cell. (1) However, I believe the zwittergent removes attached CPS. If this information is true of the steps performed, that the bulk of the CPS was discarded, and I'm not confident that what is reported is close to an accurate measure of true CPS production. The units reported differ from 2 lab-to-lab, so I cannot evaluate if these are in line with other publications. (2) As there is a significant amount of CPS that is released into the supernatant, I'm not sure that the stated goal of measuring only cell-attached CPS is the most accurate representation of CPS production. While I do not think these particular data will alter the other findings reported here, I do feel this needs to be addressed. The authors should clarify their protocol steps and justify why measuring cell-associated CPS was chosen over total CPS.

9. A separate concern about the CPS quantification is the apparent calculations. Line 602, it is stated that OD520 was divided by OD600. The normalization step should be performed after OD520 is used to calculate CPS concentration. If the authors have done the math both ways and show that it does not matter, I would be satisfied, but might suggest they rephrase more simply that CPS production was normalized to culture OD to avoid raising concerns from other readers. I place this in the "major" concerns section as it needs to be resolved/clarified for accurate reporting, but I can see that it likely will not change interpretation of the data.

10. I have another procedural question regarding the HMV assay. Line 616 states the cultures were spun at 2500 rpm. This is not an ideal unit as rpm-to-centripetal force varies between centrifuges. Please report this as "# x g" to allow comparison with other studies. Also, please clarify "background noise was subtracted" (line 618), and if the cultures were normalized to a constant OD prior to centrifugation.

Minor Comments

1. Lines 83 & 90, I understand the need for simplification, but I am concerned that use of the term "serotypes" to describe the different capsule types since not all have actually been serotyped and assigned a K type designation based on sequencing alone. If this is only to refer to the K types in this study, it would be acceptable but this should be stated a bit more clearly.

2. Fig. 1A needs definitions/symbols to identify the sugar precursors & color scheme in the heat map. In figs. 1B&C, it seems odd to me to present these data as a line graph because each has different variable on the X-axis, rather that a single variable (e.g. time, concentration). I wonder if stacked bar graphs wouldn't be more appropriate for conveying the key points?

3. Fig S4, the scale is very important in this figure as it varies in each plot and needs to be easily visualized

4. Fig S5, 52.145-K3 has an unusual growth pattern—is this reproducible?

5. Lines 310-311, the logic for performing the in vitro evolution experiment feels countered by the absence of a fitness cost from the growth curves. I do think this is an important experiment with interesting findings, but I would suggest setting up the justification a bit differently to avoid others being confused (as I was).

6. Lines 349-350, Mike, et al 2020 (reference #38) would be appropriate to include here.

7. Several times, the phrase "precultures were grown overday". I do not understand what this means and would appreciate some clarity. Were colonies inoculated in the morning, then subcultured late in the day? Were these subcultures of overnight cultures?

8. The method for bile salt survival seems a bit unusual—is there a reference for this approach? More typically, there is an incubation with the target agent for a period of time, then dilution and plating rather than plating on solid medium with the agent.

9. In the section describing introduction of the mCherry gene (beginning line 652), as written, some terminology is incorrect. It reads to me that this is an insertion event via homologous recombination within the glmS gene, rather than specific integration at the Tn7 site. Please clarify by stating more precisely where the gene was integrated.

10. The statement beginning on line 388 that K1 increased HMV is consistent with the observations in Salisbury, et al. (PMID: 41363447), however the contrasting point that K3 reduced HMV is very intriguing. Can the authors speculate on why this may be? Are there notable differences that might contribute to this? I think some additional attention to differences observed between rmp+ and rmp- host strains might also be informative.

11. Line 352, Walker, et al. 2020 (PMID: 32963003) would be a more appropriate reference or additional reference to include here.

---

## [Decision Letter · Decision Letter 2]

5 Mar 2026

Dear Mrs Le Bris, dear Olaya,

Thank you for your patience while we considered your revised manuscript "Capsule type swapping in Klebsiella spp. by plug-and-play" for publication as a Research Article at PLOS Biology. This revised version of your manuscript has been evaluated by the PLOS Biology editors, the Academic Editor and the original reviewers.

Based on the reviews, we are likely to accept this manuscript for publication, provided you satisfactorily address the remaining editorial points. Please also make sure to address the following data and other policy-related requests.

1) We routinely suggest changes to titles to ensure maximum accessibility for a broad, non-specialist readership, and to ensure they reflect the contents of the paper. In this case, we would suggest a minor edit to the title, as follows. Please ensure you change both the manuscript file and the online submission system, as they need to match for final acceptance:

“Plug-and-play evolution of the Klebsiella pneumoniae capsule locus enables serotype exchange across genetic backgrounds”

2) Thank you for providing the financial statement. However, we notice the statements are different from the statement provided within the manuscript to that in the Manuscript Details (when you submit). Could you please update whichever is the correct one? Please also add the weblink of the funding agencies in the Financial Disclosure statement in the manuscript details.

Please supply the numerical values either in the a supplementary file or as a permanent DOI’d deposition for the following figures:

Figure 1B, 2AB, 3A-D, 4A-D, 5A-D, S2ABC, S3BC, S4, S5, S6ABC, S7AB, S8A-D, S9AB, S10ABC

4) Please cite the location of the data clearly in all relevant main and supplementary Figure legends, e.g. “The data underlying this Figure can be found in S1 Data” or “The data underlying this Figure can be found in https://doi.org/10.5281/zenodo.XXXXX”

5) Supplementary files (e.g., excel). Please ensure that all data files are uploaded as 'Supporting Information' and are invariably referred to (in the manuscript, figure legends, and the Description field when uploading your files) using the following format verbatim: S1 Data, S2 Data, etc. Multiple panels of a single or even several figures can be included as multiple sheets in one excel file that is saved using exactly the following convention: S1_Data.xlsx (using an underscore).

6) Please ensure that your Data Statement in the submission system accurately describes where your data can be found and is in final format, as it will be published as written there

7) Per journal policy, if you have generated any custom code during the course of this investigation, please make it available without restrictions. Please ensure that the code is sufficiently well documented and reusable, and that your Data Statement in the Editorial Manager submission system accurately describes where your code can be found. More information on our Code Policy, what and how to share can be found here: https://journals.plos.org/plosbiology/s/code-availability

We expect to receive your revised manuscript within two weeks.

*Published Peer Review History*

*Press*

Sincerely,

Melissa

Melissa Vazquez Hernandez, Ph.D.

Associate Editor

PLOS Biology

REVIEWERS' COMMENTS

Reviewer #2 (Alvaro San Millan): The authors have adressed all my comments. Congratulations.

Reviewer #3: This revised manuscript by Le Bris, et al. is much improved! I am satisfied with the authors' responses to my concerns, as well as how they addressed those expressed by the other reviewers. The additional data has added clarity, and I think make this an even more interesting story.

---

## [Editor Report · Decision Letter 3]

9 Mar 2026

Dear Dr Le Bris,

Thank you for the submission of your revised Research Article "Plug-and-play evolution of the Klebsiella pneumoniae capsule locus enables serotype exchange across genetic backgrounds" for publication in PLOS Biology. On behalf of my colleagues and the Academic Editor, Arjan de Visser, I am pleased to say that we can in principle accept your manuscript for publication, provided you address any remaining formatting and reporting issues. These will be detailed in an email you should receive within 2-3 business days from our colleagues in the journal operations team; no action is required from you until then. Please note that we will not be able to formally accept your manuscript and schedule it for publication until you have completed any requested changes.

PRESS

Sincerely,

Melissa

Melissa Vazquez Hernandez, Ph.D., Ph.D.

Associate Editor

PLOS Biology
